# Unleashing LLMs in Bayesian Optimization: Preference-Guided Framework for Scientific Discovery

**Xinzhe Yuan**[2,1,†]    **Zhuo Chen**[1,3,†]    **Jianshu Zhang**[1,3]

Huan Xiong[2,‡]    Nanyang Ye[3,‡]    Yuqiang Li[1,‡]    Qinying Gu[1,‡]

† Equal contribution.

‡Corresponding authors: {`huan.xiong.math,ynylincolncam`}`@gmail.com`,
                        {`liyuqiang,guqinying`}`@pjlab.org.cn`

[1]Shanghai Artificial Intelligence Laboratory, Shanghai, China
[2]Harbin Institute of Technology, Harbin, China
[3]Shanghai Jiao Tong University, Shanghai, China

## Abstract

Scientific discovery is increasingly constrained by costly experiments and limited resources, underscoring the need for efficient optimization in AI for science. Bayesian Optimization (BO), though widely adopted for balancing exploration and exploitation, often exhibits slow cold-start performance and poor scalability in high-dimensional settings, limiting its applicability in real-world scientific problems. To overcome these challenges, we propose LLM-Guided Bayesian Optimization (LGBO), the first LLM preference-guided BO framework that continuously integrates the semantic reasoning of large language models (LLMs) into the optimization loop. Unlike prior works that use LLMs only for warm-start initialization or candidate generation, LGBO introduces a region-lifted preference mechanism that embeds LLM-driven preferences into every iteration, shifting the surrogate mean in a stable and controllable way. Theoretically, we prove that LGBO does not perform significantly worse than standard BO in the worst case, while achieving significantly faster convergence when preferences align with the objective. Empirically, LGBO consistently outperforms existing methods across diverse dry benchmarks in physics, chemistry, biology, and materials science. Most notably, in a new wet-lab optimization of Fe-Cr battery electrolytes, LGBO attains **90% of the best observed value within 6 iterations**, whereas standard BO and existing LLM-augmented baselines require more than 10. Together, these results suggest that LGBO offers a promising direction for integrating LLMs into scientific optimization workflows.

## 1 Introduction

Traditional experimental approaches have long driven scientific discovery. However, their labor-intensive and time-consuming nature continues to limit the pace at which new insights can emerge (Xie et al., 2023; Tom et al., 2024; Seifrid et al., 2022). To alleviate these bottlenecks, self-driving laboratories combine robotic execution with artificial intelligence, enabling systematic and high-throughput exploration (Chen et al., 2024; Ai et al., 2024). Within these platforms, Bayesian Optimization (BO) has emerged as a powerful strategy, balancing exploration and exploitation while offering principled uncertainty quantification (Guo et al., 2023; Shields et al., 2021; Shahriari et al., 2015).

BO has achieved success across diverse scientific domains, but it faces two well-known challenges. First, the scarcity of early experimental data makes cold-start optimization slow and inefficient. Second, high-dimensional parameter spaces introduce the curse of dimensionality, further limiting scalability (Guo et al., 2023). These issues often hinder BO in precisely the settings where it is most

needed-exploratory, expensive scientific tasks. Several acceleration strategies have been proposed, including transfer learning (Chowdhury & Gopalan, 2021), embedding methods (Nayebi et al., 2019; Moriconi et al., 2020), and adaptive partitioning (Wang et al., 2020). Yet, none of these methods fully addresses the cold-start bottleneck.

Recent efforts have attempted to bring large language models (LLMs) into BO. A first line of work uses LLMs for *warm-up*, letting the model propose the initial design points before the optimization loop begins. This strategy can provide a strong prior when the task is well aligned with the model's knowledge, accelerating the otherwise slow cold-start phase (Liu et al.). A second line of work leverages LLMs to *propose candidate points* during the loop, which are then re-ranked or filtered by a standard acquisition function (Chang et al., 2025; Yin et al., 2024; Yang et al., 2025). These approaches demonstrate the promise of linguistic priors, but they also share a critical limitation: the LLM's guidance remains only loosely embedded in the optimization loop—either injected once at initialization or later overridden by the acquisition function. This motivates a central question:

**Can we design a BO framework that more fully integrates LLM preferences throughout the optimization process, rather than confining them to warm-up initialization or auxiliary candidate proposals?**

To this end, we present the *LLM-Guided Bayesian Optimization (LGBO) framework*. Rather than restricting LLMs to one-off suggestions, LGBO treats them as expert priors that provide preferences and directional guidance throughout the acquisition process. This allows the optimizer to leverage LLM reasoning when informative, while remaining robust to weak or noisy signals.

We also evaluate this framework across diverse scientific optimization tasks. Our results show that LLM guidance can substantially accelerate BO when informative, while performance remains competitive under weaker signals. Furthermore, we provide theoretical guarantees ensuring the consistency of the framework. Our contributions are as follows:

1. We propose an *LLM-preference BO* framework that incorporates LLM expertise into Bayesian optimization through continuous preference guidance, moving beyond existing warm-start or candidate-proposal paradigms.

2. We provide theoretical guarantees showing that aligned LLM guidance can substantially accelerate convergence, while even under misleading guidance the additional cost remains provably bounded.

3. We validate the framework on a broad suite of scientific optimization tasks, spanning diverse dry benchmarks and a wet-lab electrolyte optimization experiment, demonstrating both generality and practical utility.

## 2 RELATED WORK

In this section, we review existing studies and analyze their limitations. We begin with approaches that incorporate large language models into Bayesian optimization, then turn to human-in-the-loop methods where expert preferences guide the search, and finally discuss how these two lines connect to our framework.

### 2.1 LLMS IN BAYESIAN OPTIMIZATION

Before reviewing how LLMs have been integrated, it is helpful to first recall what BO is. BO provides a way to search for the best solution when directly testing the target is expensive and time-consuming. The idea is to treat the true objective as a "black box": we can try different inputs and observe the outcomes, but we do not know its exact formula. To make the search more efficient, BO builds a surrogate model—often a Gaussian Process (GP)—that approximates the black box based on the points already tested. At each step, this surrogate is updated with new observations, and the acquisition function, defined on the surrogate, proposes the next point to evaluate. The new outcome is then obtained from the real black box, after which the surrogate is updated and the cycle repeats.

Recently, studies have explored using large language models to assist this BO process. LLAMBO (Liu et al.) leverages LLMs to warm-start the optimization with informative initial points and to generate candidate samples conditioned on past evaluations. However, the final decision at each iteration

is still made by a standard acquisition function over a surrogate, leaving the LLM's reasoning only indirectly involved. Subsequent studies explore similar ideas in domain-specific settings. ADO-LLM (Yin et al., 2024) applied LLM-assisted proposals to analog circuit design, and ReasoningBO (Yang et al., 2025) highlighted how contextual reasoning can enrich candidate generation. Most recently, LLINBO (Chang et al., 2025) proposed a more systematic "LLM-in-the-loop" framework, designed for the early exploration stage of BO to address the cold-start problem. In addition, beyond the scientific discovery domain, several LLM–BO approaches have also emerged in other areas. For instance, BOPRO (Agarwal et al., 2025) is a method that leverages large language models as implicit Bayesian priors through in-context learning. CAKE (Suwandi et al., 2025), in contrast, utilizes LLMs to construct kernel functions tailored to specific tasks, enabling structured and data-efficient optimization over complex input spaces.

Across these frameworks, however, LLMs are treated as auxiliary components rather than integral parts of the loop, with their guidance incorporated only partially and indirectly. This gap motivates the development of approaches that continuously and systematically embed evolving model preferences into the optimization loop.

## 2.2 HUMAN EXPERTS IN BAYESIAN OPTIMIZATION

A natural source of inspiration is human-in-the-loop Bayesian optimization, where algorithms exploit the fact that experts often find it easier to provide preferences or qualitative guidance rather than precise numerical labels. Early work on preference-based BO formalized how such comparative judgments can be integrated into acquisition strategies (Brochu et al., 2010; González et al., 2017). More recent systems extend this idea by modeling expert reliability (Xu et al., 2024) or progressively eliciting knowledge through targeted queries (Huang et al., 2022), showing that expert feedback can substantially improve sample efficiency even when partial or noisy.

ColaBO (Hvarfner et al., 2024) provides a general framework in which experts specify a preference, and the optimizer reweights sampled paths based on how well the best candidates identified along those paths align with that preference. This design is effective with human experts, whose input is typically stable and given once at initialization. With LLMs as experts, however, two challenges emerge. First, since ColaBO accepts only an initial preference, it cannot naturally incorporate updated guidance during the optimization process, leaving the LLM's capacity for ongoing reasoning underutilized. Second, if preferences are revised over time, the required reweighting of sampled paths would need to be updated continually, which may introduce instability or even divergence. These factors make ColaBO less directly applicable to LLM-guided BO and point to the need for frameworks that can embed evolving model preferences in a stable manner.

## 3 LLM-GUIDED BAYESIAN OPTIMIZATION

We present *LLM-Guided Bayesian Optimization*, a framework that integrates large language models into Bayesian optimization in a stable and tractable way. The key challenge is to incorporate external preferences without sacrificing mathematical rigor: standard BO relies purely on data-driven posteriors, while preference-based extensions for human experts assume sparse, stable, or one-off feedback. LLM guidance, by contrast, is iterative, coarse, and potentially noisy. Directly injecting such signals into the acquisition function can destabilize the optimization loop.

LGBO addresses this challenge through the *region-lifted preference*, a mechanism that translates LLM suggestions into lightweight priors which shift the surrogate mean but leave its covariance unchanged. This design allows semantic guidance to be incorporated continuously while preserving the statistical properties of BO. Section 3.1 provides the preliminaries of standard BO and prior expert-guided BO, Section 3.2 introduces Region-Lifted Preference as the core part of the LGBO framework, Section 3.3 details the optimization loop, and Section 3.4 presents theoretical guarantees.

## 3.1 PRELIMINARIES

We first recall the basic setup of Bayesian optimization and the challenges of incorporating external preferences.

**Standard BO.** Given observed data $D_t = \{(x_i, y_i)\}_{i=1}^t$, BO maintains a posterior distribution over the objective $f : \mathcal{X} \to \mathbb{R}$, typically modeled with a Gaussian process $\mathrm{GP}(\mu, k)$, where $\mu$ is mean function and $k$ is kernel (covariance) function. This posterior provides both mean predictions and uncertainty estimates. An acquisition function $a_t(x)$ (e.g., EI or UCB) then selects the next query by balancing exploration and exploitation to efficiently approach the global optimum (Shahriari et al., 2015).

**Preference as a functional.** Most expert-guided Bayesian optimization methods can be viewed through a preference functional $\rho(f)$, which reweights the posterior toward functions that better match external guidance. For example, a statement like "$x_1$ is preferred to $x_2$" can be modeled as $\rho(f) = \sigma(f(x_1) - f(x_2))$, where $\sigma$ is the sigmoid function. The standard BO posterior $p(f \mid D_t)$ represents our belief about the black-box function after observing data $D_t$, and multiplying it with $\rho(f)$ produces an adjusted posterior

$$p(f \mid D_t, \rho) \propto p(f \mid D_t)\, \rho(f),$$

which incorporates additional preferences directly into the BO loop. In practice, however, such posteriors are often intractable, especially when $\rho(f)$ depends on global properties of $f$. For this reason, existing human-in-the-loop optimization methods usually inject preferences indirectly by modifying the acquisition function. While this strategy is effective under stable human feedback, letting LLMs directly alter the acquisition function is risky, as it can cause instability or even divergence of the optimization process.

## 3.2 Core Tool: Region-Lifted Preference

We argue that LLMs are naturally better at providing *coarse, semantic guidance*—such as identifying promising regions (e.g., "try near high temperature and low pressure")—rather than precise pairwise comparisons. To leverage this strength while preserving tractability, we introduce a *region-lifted preference* scheme: the LLM specifies a preferred region $R \subseteq \mathcal{X}$, and the preference functional is defined as a linear exponential lift over a discretization $\mathcal{G} = \{x_g\}_{g=1}^G \subset R$:

$$\rho(f) = \exp\Big(\lambda \sum_{g=1}^G a_g f(x_g)\Big) \; = \; \exp\big(\lambda\, a^\top F_\mathcal{G}\big),$$

where $a = [a_1, \dots, a_G]^\top \in \mathbb{R}^G$ with $a_g \geq 0$ (e.g., uniform $a_g = 1/G$), $F_\mathcal{G} = [f(x_1), \dots, f(x_G)]^\top \in \mathbb{R}^G$, and $\lambda > 0$ controls the guidance strength. This functional encourages higher function values in $R$ without explicitly modeling optimality, thereby avoiding the intractability of argmax-dependent preferences.

Although this regional lift appears complex, proposition 1 shows that it can be implemented exactly and tractably, as a mean shift of the GP prior, preserving the Gaussian structure and enabling seamless integration into standard BO pipelines.

**Proposition 1** (Exponential lift equals mean shift)**.** *Let $f \sim \mathrm{GP}(\mu, k)$ and consider a linear functional over a grid $\mathcal{G} = \{x_g\}_{g=1}^G$ with weights $a \in \mathbb{R}^G$. Lifting the posterior distribution by*

$$\exp\big(\lambda\, a^\top F_\mathcal{G}\big)$$

*yields a new Gaussian process with the same covariance kernel $k$, but a shifted mean function*

$$\mu_\lambda(x) \; = \; \mu(x) + \lambda \sum_{g=1}^G a_g\, k(x, x_g).$$

*Equivalently, for any finite query set $X$, the marginal distribution is*

$$F_X \sim \mathcal{N}\big(\mu_X + \lambda\, \Sigma_{X\mathcal{G}} a, \; \Sigma_{XX}\big).$$

*In particular, the expected regional lift is*

$$\Delta = \mathbb{E}_\lambda[a^\top F_\mathcal{G}] - \mathbb{E}[a^\top F_\mathcal{G}] = \lambda\, a^\top \Sigma_{\mathcal{G}\mathcal{G}} a.$$

Here $\mathcal{N}(\mu, \Sigma)$ denotes the multivariate normal distribution. This result is a direct finite-dimensional consequence of Theorem 2 in Appendix A, which establishes the general Cameron–Martin form of exponential lifting for Gaussian measures.

**Region-lifted as a tool.** Building on Proposition 1, the region-lifted preference provides a practical interface for large language models. In the system prompt, the model's output is strictly constrained to one of two formats:

1) **Point mode:** `[point, [x1, x2, ..., xd], ccc]`,

2) **Region mode:** `[region, [[lb1, ..., lbd], [ub1, ..., ubd]], ccc]`.

Here, `ccc` is a confidence score in $[0, 1]$. In region mode, the model specifies a hyper-rectangle through `lb`/`ub`; we discretize it using a Sobol grid $\mathcal{G}$. In point mode, the lift is localized: we construct $a$ as a smoothly decaying weight vector centered at the suggested point, so that $a^\top F_{\mathcal{G}}$ captures a soft neighborhood average. This structured interface allows the LLM to express coarse semantic preferences while ensuring that its influence remains mathematically tractable.

**Calibrating the guidance strength.** To ensure that the lift is well-scaled, we map the LLM's confidence $c \in [0, 1]$ directly to the guidance strength. From Proposition 1, under exponential tilting, the effect of a lift is equivalent to shifting the regional functional $a^\top F_{\mathcal{G}}$ by

$$\Delta(\lambda) = \lambda \, a^\top \Sigma_{\mathcal{G}\mathcal{G}} a,$$

so the choice of $\lambda$ directly determines the effective magnitude of this shift. We set

$$\lambda = \frac{c}{\sqrt{a^\top \Sigma_{\mathcal{G}\mathcal{G}} a}},$$

where $\Sigma_{\mathcal{G}\mathcal{G}}$ is the GP posterior covariance on $\mathcal{G}$, and $a^\top \Sigma_{\mathcal{G}\mathcal{G}} a$ is the posterior variance of the regional functional. This normalization calibrates the lift to a consistent "$c$-standard-deviation" scale—regions with low posterior uncertainty receive a small effective shift, while uncertain regions receive a larger one. In this way, the LLM's preference integrates smoothly with the model's uncertainty without introducing extra hyperparameters or risking premature over-exploitation.

### 3.3 LGBO Framework

LGBO integrates these region-lifted preferences into BO in a continuous loop. Instead of altering the acquisition function directly, LLM outputs reshape the surrogate model at each iteration. This maintains robustness while allowing semantic guidance to evolve alongside experimental data.

**From warm-start to continuous guidance.** Earlier approaches have used LLMs in limited ways: either to provide one-off initialization or auxiliary candidate proposals that are later filtered by the acquisition function. In both cases, the guidance is only loosely connected to the optimization loop. LGBO departs from these approaches by embedding LLM preferences at every iteration. At every iteration, the surrogate is informed not only by newly collected experimental data but also by updated LLM preferences, which are systematically incorporated through region-lifted priors. This design allows semantic signals, such as domain heuristics or qualitative trends, to shape the surrogate throughout the process, ensuring that LLM reasoning evolves in tandem with data and becomes a persistent part of the optimization loop.

**Workflow.** The LGBO loop, illustrated in Figure 1, follows the structure of standard BO with one key modification: region-lifted updates are injected at each iteration. The process begins with initialization, where the LLM provides starting points based on prior knowledge. After each evaluation, the GP surrogate is retrained, and the LLM is queried with a structured prompt with fixed components (domain knowledge, constraints, and the required output format) and a dynamic user prompt summarizing experimental history plus the reasoning trace from the previous round. The model then outputs either a point or a region with a confidence score, which is converted into a region-lifted preference and applied as a mean shift of the surrogate mean while leaving the covariance unchanged. The acquisition function, such as EI or UCB, then selects the next experiment based on this updated surrogate. By repeating this cycle, LGBO integrates semantic guidance continuously, while the acquisition function retains full control over decision-making. The full prompt specification is provided in Appendix B.

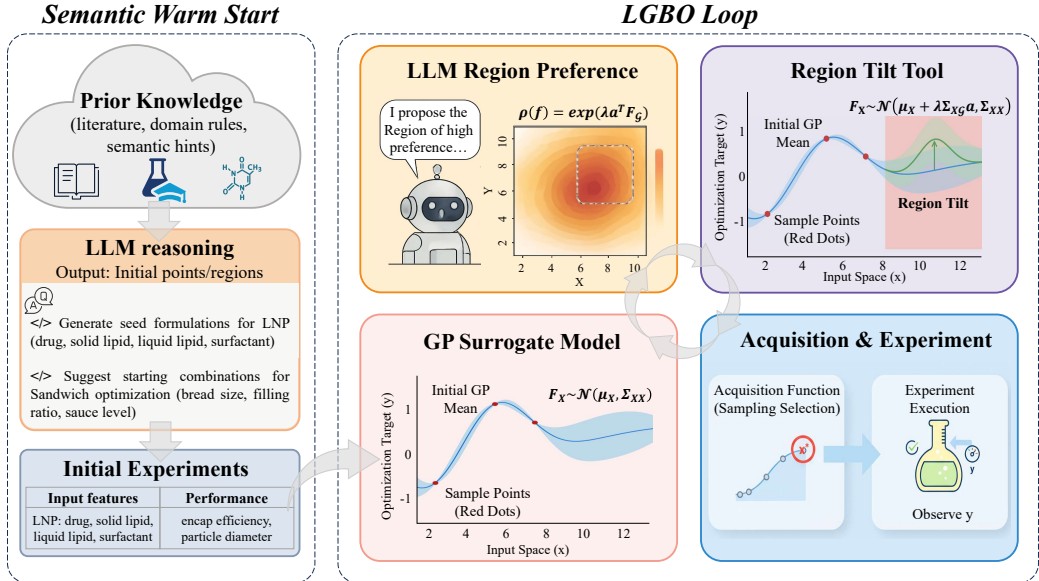

Figure 1: LGBO framework. The proposed LLM-Guided Bayesian Optimization integrates prior knowledge and LLM reasoning to warm-start the search, and iteratively combines a GP surrogate with region-lifted preferences. At each round, the LLM provides coarse regional guidance, implemented as a mean shift of the surrogate, while the acquisition function selects the next experiment. This design ensures stability in the worst case while enabling acceleration when the guidance is informative.

**Advantages of semantic guidance.** This design allows LGBO to harness aspects of LLM reasoning that are otherwise inaccessible to purely data-driven methods. For instance, the model can highlight regions suggested by chemical intuition (e.g. higher temperature and lower concentration may accelerate reaction yield) or by cross-domain analogies (e.g., drawing parallels between material compositions). By embedding such regional signals into the surrogate, LGBO guides exploration toward scientifically meaningful areas without abandoning the statistical rigor of BO.

**Guarantees.** Our framework provides two complementary guarantees: in the worst case of irrelevant or misleading guidance, the induced lift only enlarges the surrogate's norm bound, so regret remains within a constant factor of standard BO; when guidance aligns with the true objective, the effective radius shrinks, yielding strictly tighter regret bounds (see Section 3.4). Thus LGBO is provably safe, yet can substantially accelerate convergence when preferences are informative.

## 3.4 THEORETICAL GUARANTEE

We now provide regret guarantees for LGBO. In the full LGBO algorithm, the LLM proposes a new region at every BO iteration, producing an adaptive sequence of lifts $\{\tau_t\}$. The theoretical analysis is not intended to characterize this fully dynamic process. Instead, we study a simplified but mathematically tractable variant in which a single preference direction $g$ is fixed at the beginning of optimization, and the corresponding lift $\tau(x) = \lambda g(x)$ is applied consistently throughout the $T$ rounds.

This "frozen-lift" setting is a reasonable abstraction because, in scientific optimization tasks, a domain-informed LLM often exhibits structurally coherent behavior: once the model identifies features associated with near-optimal or high-quality regions, its suggested regions tend to remain clustered around similar areas rather than fluctuating dramatically across iterations. A fixed direction therefore serves as a representative surrogate for a stable class of regional preferences while avoiding the technical complications of analyzing a fully adaptive sequence.

Under this setting, the lifted objective $f' = f - \tau$ remains a single element of the RKHS, allowing direct application of classical GP-UCB theory. The analysis isolates the mechanism-level effect of one regional preference direction: when the direction is aligned with the true objective, the effective RKHS radius contracts and the regret bound becomes strictly tighter; when the direction is misaligned, the radius increases only by a constant amount, and the worst-case rate matches standard GP–UCB. Thus, although the fixed-lift variant is not an exact description of the full LGBO dynamics, it provides a principled and verifiable abstraction of the core preference mechanism.

**Lift representation.** Given a finite grid $\{x_i\}$ and non-negative weights $a_i$, the structured region or point suggested by the LLM is mapped, under the LGBO lifting rule defined in Section 3.2, to the kernel-induced function

$$g(x) = \sum_i a_i k(x, x_i), \qquad \tau(x) = \lambda g(x).$$

This yields an equivalent mean-shifted representation of the prior as $\mathrm{GP}(\mu + \tau, k)$.

**Regularity assumptions.** We follow the classical GP–UCB assumptions (Srinivas et al., 2009):

$$f - \mu \in \mathcal{H}_k, \qquad \|f - \mu\|_{\mathcal{H}_k} \leq B_0,$$

where $\|\cdot\|_{\mathcal{H}_k}$ denotes the RKHS norm associated with kernel $k$ and $R$-sub-Gaussian noise.

**Alignment coefficient.** The analysis uses an alignment coefficient

$$c := \frac{\langle f - \tau, g \rangle_{\mathcal{H}_k}}{\|f - \tau\|_{\mathcal{H}_k} \|g\|_{\mathcal{H}_k}} \in [-1, 1],$$

which measures how well the lifted direction $g$ agrees with the centered objective $f - \mu$ in the RKHS. Building on the fixed lift and the alignment structure introduced above, we can now state the corresponding regret bounds.

**Theorem 1** (Lifted regret bounds). *If $f \sim \mathrm{GP}(\mu, k)$ satisfies $\|f - \mu\|_{\mathcal{H}_k} \leq B_0$, then under the fixed lift $\tau(x) = \lambda g(x)$ defined above. Run GP-UCB on the residual labels $y'_t = y_t - \tau(x_t)$ with confidence parameter $\beta_T(\cdot)$ and let $\gamma_T$ denote the maximal information gain. Then, with probability at least $1 - \delta$,*

$$R_T \leq \sqrt{8T \, \beta_T(B_{\mathrm{out}}) \, \gamma_T}, \qquad B_{\mathrm{out}} = B_0 + \lambda \|g\|_{\mathcal{H}_k},$$

*and, when the alignment coefficient satisfies $c > 0$,*

$$R_T \leq \sqrt{8T \, \beta_T(B_{\mathrm{in}}) \, \gamma_T}, \qquad B_{\mathrm{in}} = B_0 \sqrt{1 - c^2}.$$

*All hidden constants are universal. Proof is given in Appendix A.*

**Remark 1** (Reading the bounds). *For standard BO, the complexity is controlled by $B_0$, yielding regret on the order of $\sqrt{8T \, \beta_T(B_0) \, \gamma_T}$ (Srinivas et al., 2009). Under LGBO, the misaligned case simply replaces $B_0$ by $B_{\mathrm{out}} = B_0 + \lambda \|g\|_{\mathcal{H}_k}$, so the bound remains comparable up to a lift-dependent cost. In the aligned case, the effective radius shrinks to $B_{\mathrm{in}} = B_0 \sqrt{1 - c^2}$, which can be much smaller than $B_0$ whenever $c$ is large, leading to strictly tighter regret. Thus LGBO is not perform significantly worse than standard BO, and can be substantially better when the suggested region is informative.*

## 4 EXPERIMENTS

We evaluate LGBO in two settings: dry experiments, which use pre-collected public datasets, and a wet experiment conducted in a real laboratory. Dry experiments provide a finite set of design-response pairs, from which an offline optimum can be identified. While this optimum may not coincide with the true global optimum of the underlying physical system, it serves as a consistent reference for comparing algorithms. In contrast, the wet experiment involves an ongoing task with limited data, measurement noise, and an unknown optimum, representing a realistic scenario where optimization must proceed under laboratory constraints.

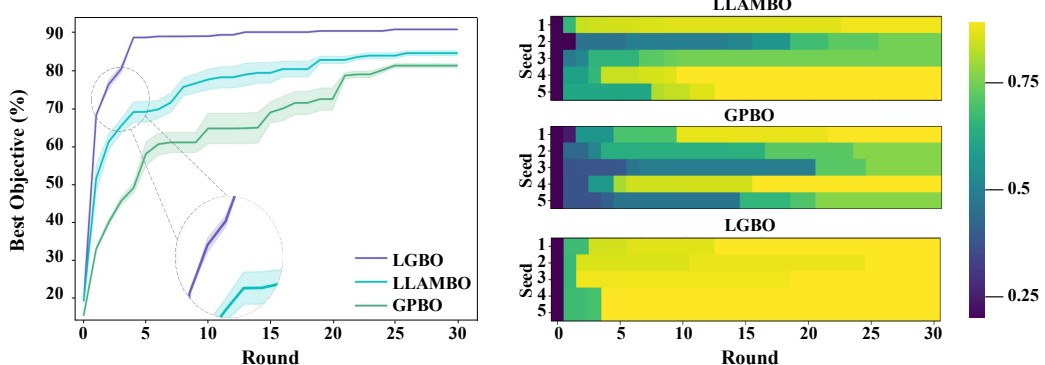

Figure 2: LNP3 results. **Left:** LGBO achieves faster convergence and higher final performance than GPBO and LLAMBO. **Right:** trajectory heatmaps show that LGBO converges more rapidly and consistently across seeds.

## 4.1 EXPERIMENT SETTING

**Baselines.** Two representative methods are used for comparison: (i) **GPBO**, the canonical Gaussian-process Bayesian optimization framework with Matérn-5/2 kernel; (ii) **LLAMBO** (Liu et al.), a state-of-the-art LLM-augmented BO approach that leverages language models for warm-start initialization and candidate generation.

**Implementation details.** All methods use the same surrogate model (GP with Matérn-5/2 kernel), and the log-$q$EI acquisition function. GP hyperparameters are re-optimized after each iteration via marginal likelihood. Each run begins with two initial evaluations: **GPBO** uses Sobol initialization, while **LLAMBO** and **LGBO** share the same LLM-suggested points for fairness. Results are reported as mean $\pm$ variance deviation over five seeds. For LLM-based methods, we use **Intern-S1-241B**, a pretrained large model in the scientific domain (Bai et al., 2025). Prompts are designed so that no task-specific objectives or seeds are exposed, ensuring that the LLM cannot rely on memorized knowledge.

**Dry experiment datasets.** Four benchmarks are used to assess generality across scientific domains. Detailed specifications are given in Appendix C:

1) **LNP3 (Chen et al., 2022):** lipid nanoparticle formulations for cannabidiol delivery, optimizing for drug loading, encapsulation efficiency, and particle size. The search space has five design variables: four mixture inputs (three continuous, one discrete) and one categorical lipid type.

2) **Cross-barrel (Gongora et al., 2020):** 3D-printed crossed-barrel structures, optimized for mechanical toughness. The design variables are four continuous geometry parameters $(n, \theta, r, t)$.

3) **Concrete (Tripathi, 2020):** mixture design of cement, aggregates, water, and admixtures, with the objective of maximizing compressive strength under realistic composition constraints.

4) **HPLC (Häse et al., 2021):** automated high-performance liquid chromatography, where six process parameters (e.g., loop size, flow rate, pump settings) are tuned to maximize chromatographic peak area.

**Fe-Cr redox flow battery (wet experiment).** The wet experiment focuses on optimizing electrolyte formulations for Fe-Cr redox flow batteries, a system for which no benchmark dataset exists and where the true optimum is unknown. Thus, LLMs cannot rely on memorized training data and must instead draw on general scientific knowledge. Key ion concentrations (HCl, $FeCl_{2/3}$, $CrCl_{3/2}$) jointly determine viscosity and conductivity, and all measurements are aggregated into a weighted scalar objective reflecting practical application priorities. Compared to dry experiments, this setting offers few observations and is subject to laboratory noise and measurement uncertainty, providing a stringent test of whether LGBO can use qualitative priors to guide exploration under scarce data.

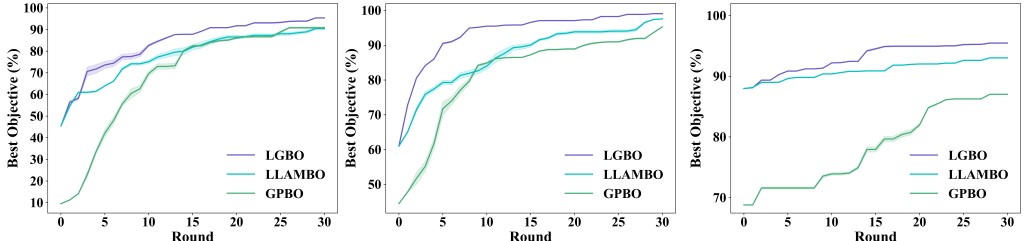

Figure 3: Convergence traces on the three dry benchmark tasks. From left to right: **HPLC**, **Cross-barrel**, and **Concrete**.

## 4.2 MAIN RESULTS

**Dry experiments.** Figure 2 shows the main results of **LNP3**. The performance traces (left) show that LGBO converges faster and achieves higher final objectives than both baselines: GPBO suffers from cold-start, while LLAMBO improves early progress but soon plateaus. The trajectory heatmaps (right) further reveal that LGBO yields much more consistent convergence across seeds, whereas GPBO and LLAMBO exhibit large variability. This stability is not accidental: by continuously incorporating domain-informed LLM preferences through region lifting, LGBO maintains coherent search directions and avoids uninformative exploration. For completeness, Appendix D.1 provides pairplots of sampled points, showing that LGBO avoids large regions of uninformative exploration, consistent with its reduced variance and faster convergence.

Across the other benchmarks (Figure 3), LGBO consistently achieves stronger performance. On **HPLC**, the task is highly noisy and all methods fluctuate considerably, yet LGBO still maintains the highest final performance. On **Cross-barrel**, the low-dimensional design space allows GPBO to perform reasonably well, yet LGBO achieves the best toughness and explores more effectively around the optimum. On **Concrete**, both LGBO and LLAMBO benefit from LLM warm-starts, but LGBO further escapes local optima more quickly, ultimately attaining higher compressive strength. Together these results highlight the robustness and generality of LGBO across noisy, simple, and practical engineering settings.

**Wet experiment.** In the Fe-Cr electrolyte optimization (Figure 4), LGBO quickly identifies promising regions and by the sixth iteration surpasses 90% of the best observed value, while the baselines require substantially more evaluations. As optimization progresses, LGBO steadily refines its surrogate and concentrates on high-performing regions, yielding higher final performance and lower variance across runs. By contrast, GPBO and LLAMBO explore scattered, often suboptimal regions; the latter is further constrained by acquisition filtering, which prevents it from leveraging LLM guidance when the GP surrogate is crude. Together these results show that LGBO can effectively translate domain knowledge into accelerated and more reliable convergence.

## 4.3 ABLATION EXPERIENCES

To disentangle the contributions of different components in LGBO, we conduct two ablation experiments on the **HPLC** dataset, which serves as a representative benchmark beyond the main results.

**Different LLM backbones.** To evaluate robustness, we replace Intern-S1 with backbones of different sizes and capabilities: **Intern-S1-mini-7B**, as well as two larger general-purpose large language models, **Qwen3-235B-Instruct** and **Qwen3-235B-Thinking**. As shown in Figure 5, larger models exhibit stronger stability, which is further enhanced when pretrained on scientific domains. Meanwhile, **Thinking** variants tend to achieve higher final values but converge more slowly, likely because their instruction-following ability is weaker under strict prompting compared to **Instruct** models. These results demonstrate that LGBO is not tied to a specific LLM: larger or domain-pretrained models can provide richer guidance, but the framework itself remains broadly applicable across diverse backbones.

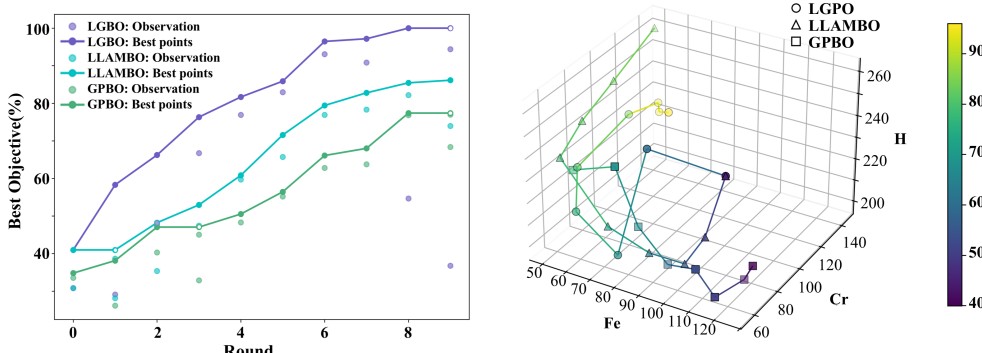

Figure 4: Wet-lab experiment results. **Left**: convergence traces across iteration rounds, showing observed values and best objective performance. The hollow markers indicate the historical best solutions observed up to that round. **Right**: 3D optimization trajectories in the chemical concentration space (Fe, Cr, additive), with color indicating the best- objective.

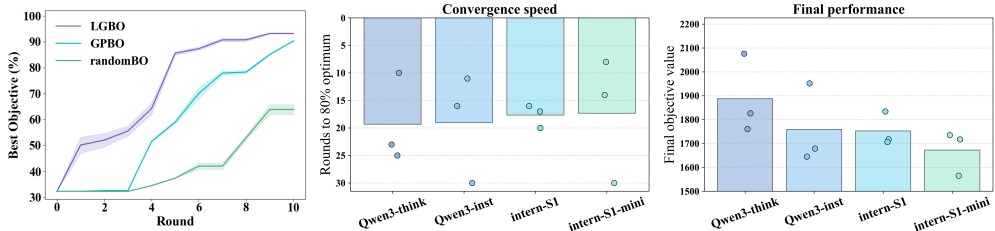

Figure 5: Ablation study results. **Left:** convergence traces comparing LGBO, GPBO, and Random region lifting BO on the HPLC task. **Middle and Right:** Different LLM backbones ablation experiment, where dots represent individual test runs and bars denote the corresponding means.

**Random region lifting.**   We replace LLM suggestions with randomly lifted regions of matched size and confidence. To eliminate warm-start effects, LGBO is also initialized from Sobol points, identical to GPBO. As shown in Figure 5, random guidance not only prevents fast convergence in the early stage but also forces the optimizer to spend more rounds on exploration. This confirms that LGBO's advantage arises from informative semantic cues provided by the LLM, rather than from the lifting mechanism alone.

**On initialization.**   Both LGBO and LLAMBO start from the same LLM-suggested initialization points, a common practice in recent literature. Yet, LLAMBO does not achieve the same acceleration as LGBO, indicating that the gain is not from initialization alone but from LGBO's continuous integration of LLM preferences.

## 5   CONCLUSION

This work introduces *LLM-Guided Bayesian Optimization*, a preference-driven framework that integrates LLMs into BO through a novel *region-lifted preference* mechanism. LGBO provides continuous guidance while preserving the rigor of BO. We establish regret bounds showing that LGBO matches BO in the worst case and accelerates convergence when guidance aligns. Ablation studies confirm that its gains stem from the continuous incorporation of LLM preferences via region lifting, and the framework is broadly applicable across different LLM backbones. On four dry benchmarks and a new wet-lab Fe-Cr electrolyte task, LGBO achieves faster convergence, lower variance, and stronger robustness than GPBO and LLAMBO, demonstrating its promise for scientific discovery.

## ACKNOWLEDGEMENTS

This work was supported by the New Generation Artificial Intelligence National Science and Technology Major Project (No. 2025ZD0121802). It was also supported by the New Generation Artificial Intelligence National Science and Technology Major Project (No. 2025ZD0122901) and the National Natural Science Foundation of China (No. 62572313).

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

## A  THEORETICAL DETAILS

**Theorem 2** (Gaussian measure under exponential linear lift). *Let $(E, \mathcal{B})$ be a separable real topological vector space, $E^*$ its continuous dual, and $\mathbb{P} = \mathcal{N}(m, C)$ a Gaussian measure with mean $m \in E$ and covariance operator $C : E^* \to E$. For any continuous linear functional $\ell \in E^*$ and any $\lambda \in \mathbb{R}$, define the lifted measure*

$$\frac{d\mathbb{P}_\lambda}{d\mathbb{P}}(f) = \exp\big(\lambda\, \ell(f) - \psi(\lambda)\big), \qquad \psi(\lambda) = \log \mathbb{E}_{\mathbb{P}}[e^{\lambda \ell(f)}].$$

*Then $f | \mathbb{P}_\lambda$ is still Gaussian with unchanged covariance $C$, and mean shifted by*

$$m_\lambda = m + \lambda C \ell.$$

*Moreover, the expected lift of the functional is*

$$\Delta = \mathbb{E}_{\mathbb{P}_\lambda}[\ell(f)] - \mathbb{E}_{\mathbb{P}}[\ell(f)] = \lambda\, \langle C\ell, \ell \rangle = \lambda\, \mathrm{Var}_{\mathbb{P}}[\ell(f)].$$

*Thus a prescribed increment $\delta$ is achievable iff $\mathrm{Var}_{\mathbb{P}}[\ell(f)] > 0$, in which case $\lambda = \Delta / \mathrm{Var}_{\mathbb{P}}[\ell(f)]$.*

*Proof.* We present the argument via finite-dimensional projections.

For any finite set of functionals $\Phi = (\phi_1, \ldots, \phi_m) \subset E^*$, the vector

$$Y = (\phi_1(f), \ldots, \phi_m(f), \ell(f))$$

is multivariate Gaussian under $\mathbb{P}$. Lifting the density by $\exp(\lambda \ell(f))$ yields, by standard Gaussian completion-of-squares or cumulant-generating function arguments, that

$$(\phi_1(f), \ldots, \phi_m(f))^\top \sim \mathcal{N}\big(\mu_\Phi + \lambda \, \mathrm{Cov}_{\mathbb{P}}((\phi_i(f))_i, \ell(f)), \; \Sigma_\Phi\big).$$

Here $\mu_\Phi = (\phi_i(m))_i$ and $\Sigma_\Phi = (\langle C\phi_i, \phi_j \rangle)_{ij}$. But

$$\mathrm{Cov}_{\mathbb{P}}(\phi(f), \ell(f)) = \langle C\ell, \phi \rangle,$$

so the new mean is exactly $(\phi_i(m + \lambda C\ell))_i$, while the covariance remains $\Sigma_\Phi$. By Kolmogorov consistency, this determines a Gaussian measure $\mathcal{N}(m + \lambda C\ell, C)$ on $E$.

Finally,

$$\Delta = \ell(m + \lambda C\ell) - \ell(m) = \lambda\, \langle C\ell, \ell \rangle,$$

which equals $\lambda\, \mathrm{Var}_{\mathbb{P}}[\ell(f)]$, yielding the desired characterization of $\lambda$. $\qquad\square$

**Lemma 1** ((Chowdhury & Gopalan, 2017, Theorem 2)). *Let $\mathcal{X}$ be compact and $k : \mathcal{X} \times \mathcal{X} \to \mathbb{R}$ a continuous kernel scaled so that $k(x, x) \leq 1$. Assume the (possibly mean-centered) target function $f$ satisfies $f \in \mathcal{H}_k$ with $\|f\|_{\mathcal{H}_k} \leq B$, and observations follow*

$$y_t = f(x_t) + \varepsilon_t, \qquad t = 1, 2, \ldots,$$

*where $\{\varepsilon_t\}$ are independent $R$-sub-Gaussian noise variables. Let $K_{t-1} = [k(x_i, x_j)]_{i,j=1}^{t-1}$, $k_{t-1}(x) = [k(x_1, x), \ldots, k(x_{t-1}, x)]^\top$, and define the GP posterior (with noise variance $\sigma^2$ and prior mean 0) by*

$$\mu_{t-1}(x) = k_{t-1}(x)^\top \big(K_{t-1} + \sigma^2 I\big)^{-1} y_{1:t-1},$$

$$s_{t-1}^2(x) = k(x, x) - k_{t-1}(x)^\top \big(K_{t-1} + \sigma^2 I\big)^{-1} k_{t-1}(x),$$

*where $y_{1:t-1} = [y_1, \ldots, y_{t-1}]^\top$. Let the (maximal) information gain be*

$$\gamma_t := \max_{A \subset \mathcal{X}, \, |A| = t} \tfrac{1}{2} \log \det\big(I + \sigma^{-2} K_A\big).$$

*Then for any $\delta \in (0, 1)$, with probability at least $1 - \delta$, simultaneously for all $t \geq 1$ and all $x \in \mathcal{X}$,*

$$\big| f(x) - \mu_{t-1}(x) \big| \leq \left( B + R \sqrt{2\big(\gamma_{t-1} + 1 + \ln \tfrac{1}{\delta}\big)} \right) s_{t-1}(x).$$

*Equivalently, defining*

$$\beta_t(B) := B^2 + 2R^2\big(\gamma_{t-1} + 1 + \ln \tfrac{1}{\delta}\big),$$

*we have*

$$big| f(x) - \mu_{t-1}(x)\big| \leq \sqrt{\beta_t(B)}\, s_{t-1}(x) \tag{1}$$

*uniformly over all $t, x$ with probability at least $1 - \delta$.*

**Lemma 2** ((Srinivas et al., 2009, Lemma 5.3 & Lemma 5.4)). *Under the same setting and notation as in Lemma 1 (with $k(x,x) \leq 1$), for any chosen sequence $\{x_t\}_{t=1}^T$ the following hold:*

1. *(Information gain decomposition)*

$$I(y_{1:T}; f) = \tfrac{1}{2} \sum_{t=1}^{T} \log\left(1 + \sigma^{-2} s_{t-1}^2(x_t)\right).$$

2. *(Capped variance sum)*

$$\sum_{t=1}^{T} \min\left\{ 1, \ \sigma^{-2} s_{t-1}^2(x_t) \right\} \leq 2\gamma_T.$$

3. *(Width-sum bound) Consequently,*

$$\sum_{t=1}^{T} s_{t-1}(x_t) \leq \sqrt{2T\gamma_T}. \tag{2}$$

**Theorem 3** (Lifted-GP-UCB: high-probability regret bound). *Let $\mathcal{X}$ be compact and $k$ continuous with $k(x,x) \leq 1$. Let the true function $f \in \mathcal{H}_k$ with $\|f\|_{\mathcal{H}_k} \leq B_0$, and observations $y_t = f(x_t) + \varepsilon_t$ where $\{\varepsilon_t\}$ are i.i.d. $R$-sub-Gaussian. Fix a grid $\mathcal{G} = \{x_g\}_{g=1}^G$ and weights $a \in \mathbb{R}^G$, and define*

$$g(x) = \sum_{g=1}^{G} a_g\, k(x, x_g), \qquad \tau(x) = \lambda\, g(x), \qquad \|g\|_{\mathcal{H}_k}^2 = a^\top \Sigma_{\mathcal{GG}} a,$$

*where $(\Sigma_{\mathcal{GG}})_{gg'} = k(x_g, x_{g'})$. Consider running GP regression on the residual labels $y'_t := y_t - \tau(x_t)$ (same kernel $k$ and noise variance $\sigma^2$), yielding posterior mean $\mu'_{t-1}$ and standard deviation $s_{t-1}$, and choose*

$$x_t \in \arg\max_{x \in \mathcal{X}} \mu'_{t-1}(x) + \sqrt{\beta_t(B)}\, s_{t-1}(x),$$

*where $\beta_t(B) = B^2 + 2R^2\left(\gamma_{t-1} + 1 + \ln\frac{1}{\delta}\right)$ and $\gamma_t$ is the maximal information gain. Define the "in-region" and conservative radii*

$$B_{\mathrm{in}} := \|f - \tau\|_{\mathcal{H}_k}, \qquad B_{\mathrm{out}} := B_0 + |\lambda|\, \|g\|_{\mathcal{H}_k}.$$

*Then, with probability at least $1 - \delta$, for all $T \geq 1$ the cumulative regret $R_T = \sum_{t=1}^{T} \left(f(x^\star) - f(x_t)\right)$ satisfies*

$$R_T \leq 2\sqrt{\beta_T(B_{\mathrm{in}})} \sum_{t=1}^{T} s_{t-1}(x_t) \leq \sqrt{8T\, \beta_T(B_{\mathrm{in}})\, \gamma_T},$$

*and, without assuming alignment,*

$$R_T \leq \sqrt{8T\, \beta_T(B_{\mathrm{out}})\, \gamma_T}.$$

*Proof.* Let $f'(x) := f(x) - \tau(x)$. By Lemma 1, for any $B \geq 0$ such that $\|f'\|_{\mathcal{H}_k} \leq B$,

$$|f'(x) - \mu'_{t-1}(x)| \leq \sqrt{\beta_t(B)}\, s_{t-1}(x), \quad \forall x \in \mathcal{X}, \ \forall t \geq 1,$$

holds with probability at least $1 - \delta$. By the selection rule,

$$f'(x^\star) \leq \mu'_{t-1}(x^\star) + \sqrt{\beta_t(B)}\, s_{t-1}(x^\star) \leq \mu'_{t-1}(x_t) + \sqrt{\beta_t(B)}\, s_{t-1}(x_t),$$

and

$$f'(x_t) \geq \mu'_{t-1}(x_t) - \sqrt{\beta_t(B)}\, s_{t-1}(x_t).$$

Thus the instantaneous regret

$$r_t = f(x^\star) - f(x_t) = f'(x^\star) - f'(x_t) \leq 2\sqrt{\beta_t(B)}\, s_{t-1}(x_t) \leq 2\sqrt{\beta_T(B)}\, s_{t-1}(x_t).$$

Summing over $t$ and applying Lemma 2 gives

$$R_T \ \le \ 2\sqrt{\beta_T(B)} \sum_{t=1}^{T} s_{t-1}(x_t) \ \le \ 2\sqrt{\beta_T(B)} \cdot \sqrt{2T\,\gamma_T} \ = \ \sqrt{8T\,\beta_T(B)\,\gamma_T}.$$

When the LLM provides unreasonable regions, we have

$$R_T \ \le \ 2\sqrt{\beta_T(B_{out})} \cdot \sqrt{2T\,\gamma_T} \ = \ \sqrt{8T\,\beta_T(B_0 + \lambda\|g\|)\,\gamma_T}.$$

Now, we focus on the bound of $B_{in}$. Let $h = f - \tau$, $c = \frac{\langle h,g\rangle}{\|h\|\|g\|}$. If LLM provides a suitable lifting area, we have $c > 0$. Let $\lambda = \frac{cB_0}{\|g\|}$ we have

$$B_{in}^2 = \|f - \tau\|_{\mathcal{H}_k} = (\lambda\|g\| - cB_0)^2 + B_0^2(1 - c^2) = B_0^2(1 - c^2).$$

So, we have

$$R_T \ \le \ 2\sqrt{\beta_T(B_{in})} \cdot \sqrt{2T\,\gamma_T} \ = \ \sqrt{8T\,\beta_T B_0\sqrt{1-c^2}\,\gamma_T}.$$

$\square$

## B  PROMPT

To ensure stability, reproducibility, and transparency, we design the prompt in a *modular* fashion, consisting of a fixed *system prompt* and a dynamic *user prompt*. The system prompt encodes general scientific principles and strict output rules that remain invariant across tasks, while the user prompt provides experiment-specific background and historical context. This separation allows LGBO to (i) maintain consistent reasoning discipline, (ii) incorporate domain knowledge explicitly, and (iii) adapt flexibly to different experimental datasets.

**User prompt.**  The user prompt augments the fixed rules with dataset-specific context. It defines the experimental background, parameter order, optimization objectives, and constraints. In addition, it incorporates both the *experimental history* (latest observations) and the *reasoning trace* from the previous round, enabling iterative refinement while avoiding overfitting to past data. To illustrate the structure, a schematic excerpt of the user prompt is shown below:

```
[Background]
- Experiment type & purpose: ...
- Parameter order (d=):...
- Objective: Minimize f(x) (single objective).
- Constraints: All parameters real-valued, dimensionless;
  respect declared order and bounds; do not normalize.
- Bounds: ...
[Review]
- Historical data (newest first):  ...
-Thinking from the previous round:...
- Adoption note: Suggestions were used as guidance;
              actual tested points may differ.
```

**System prompt.**  The system prompt frames the LLM as a scientist specializing in experimental optimization. It establishes an *evidence hierarchy*—prioritizing mechanistic background knowledge over historical data—and specifies strict output formats to ensure valid and auditable responses. The output must be either a point or a region with a confidence score, following the declared parameter order and physical units. To mitigate collapse, the prompt explicitly forbids anchoring on past best points unless mechanistically justified. Below is the core structure:

```
# Evidence hierarchy (critical)
- PRIMARY: Background knowledge, physical/chemical mechanisms,
    constraints, and units.
- SECONDARY (auxiliary only): Historical trial points/observations and
    thinking in the review.
- If background implications conflict with historical points, SIDE WITH
    BACKGROUND.

# Background (fixed across rounds)
- We run iterative chemistry experiments (e.g., polymerization/hydrolysis
    /organic synthesis).
- Parameters are physical and NOT normalized. Always use the declared
    units & order.

# Modes (pick exactly ONE)
1) [point, [x1, x2, ..., xd], ccc]
2) [region, [[lb1, lb2, ..., lbd],
             [ub1, ub2, ..., ubd]], ccc]
- ccc $\in$ [0,1].
- In region mode, each dimension must have (lb $\leq$ ub) and follow the
    declared parameter order & units.
- For categorical variables, output their literal value (e.g., "DMF") in
    both point and region.
  If a category is fixed in region, set lb=ub to that same literal value.

# How to reason (prioritize background over past points)
- Start from first principles: mechanism-driven trends, feasible/unsafe
    ranges, known monotonicities, interactions.
- Use historical data ONLY as weak corroboration or disproof of a
    background-based hypothesis.
- Do NOT anchor on previous best/nearest points; avoid proposing a point
    merely because it appeared before.
- If historical points cluster narrowly, consider a background-justified
    exploratory move (e.g., shift in a mechanism-relevant factor).
- Prefer REGION when background suggests multiple nearby settings could
    satisfy the mechanistic target; choose POINT only when background+
    data imply a sharp optimum.

# Output protocol (two blocks)
1) Thinking:
   - Be concise but informative, in this order:
     (a) Background-based rationale (mechanism/constraints) that leads to
          your proposal.
     (b) How (if at all) historical data supports/contradicts this
         mechanism ($\leq2$ sentences).
     (c) Why point vs region given the mechanism and uncertainty.
2) Final Answer:
   - Strict structure with no extra words:
     [point, [x1, x2, ..., xd], ccc]
     OR
     [region, [[lb1, lb2, ..., lbd],
               [ub1, ub2, ..., ubd]], ccc]

# Hard constraints
- Do NOT normalize or re-order parameters.
- Keep units consistent with the declared parameter order.
- No extra commentary in Final Answer beyond the bracketed structure.
# Anti-collapse checks
- Never center a region or point on a past observation unless
    mechanistically justified.
- If you reuse a past setting, explicitly state the mechanism that makes
    it optimal (in Thinking).
```

**Summary.** Together, the system and user prompts enforce a disciplined reasoning protocol: the system ensures *scientific validity and structural consistency*, while the user prompt injects *contextual knowledge and history*. This modularity makes LGBO's prompting scheme both transparent and easily transferable to new experimental domains.

## C Dry experiment datasets

We provide detailed specifications of the four dry-experiment datasets used in our study. Each dataset is summarized with variable types and ranges, and the corresponding optimization objective. Together they span formulation science, structural design, materials engineering, and process control, offering a diverse and representative testbed for Bayesian optimization.

**Lipid Nanoparticle Formulation) (Chen et al., 2022).** This dataset contains 768 samples describing lipid nanoparticle (LNP) formulations for cannabidiol (CBD) delivery. It has 5 input variables and 3 objectives. We perform 2D one-hot interpolation for solid lipid based on whether it is an acid. Because the dataset fully enumerates all variable combinations, no interpolation is required.

Table 1: LNP3 dataset specification.

| Variable | Type | Range / Options |
|---|---|---|
| drug_input | discrete | {6, 12, 24, 48} |
| solid_lipid | categorical | {Stearic_acid, Compritol_888, Glyceryl_monostearate} |
| solid_lipid_input | discrete | {120, 108, 96, 72} |
| liquid_lipid_input | discrete | {0, 12, 24, 48} |
| surfactant_input | discrete | {0, 0.0025, 0.005, 0.01} |
| drug_loading | objective | maximize |
| encapsulation_efficiency | objective | maximize |
| particle_diameter | objective | minimize |

In our experiments, each objective is normalized to $[0, 1]$ by its observed range, and the three normalized objectives are summed to define a single scalar objective.

**3D-printed structural design Cross-barrel (Gongora et al., 2020).** This dataset consists of 600 samples characterizing the performance of crossed-barrel 3D-printed structures. Interpolation (multilinear grid or kNN+IDW fallback) is applied to enable continuous evaluation.

Table 2: Cross-barrel dataset specification.

| Variable | Type | Range |
|---|---|---|
| $n$ | continuous | [6.0, 12.0] |
| $\theta$ | continuous | [0.0, 200.0] |
| $r$ | continuous | [1.5, 2.5] |
| $t$ | continuous | [0.7, 1.4] |
| toughness | objective | maximize |

**Concrete compressive strength (Tripathi, 2020).** This dataset contains 1,030 samples describing mixture design for compressive strength optimization. Age is excluded, leaving 7 continuous mixture variables. Interpolation (multilinear grid or kNN+IDW fallback) is used to build the oracle.

**HPLC (Häse et al., 2021).** This dataset contains 1,386 samples of high-performance liquid chromatography (HPLC) experiments. The objective is to maximize chromatographic peak area. Interpolation (multilinear grid or kNN+IDW fallback) is applied to extend the dataset to a continuous oracle.

Table 3: Concrete dataset specification.

| Variable | Type | Range |
|---|---|---|
| cement | continuous | [102.0, 540.0] |
| blast furnace slag | continuous | [0.0, 359.4] |
| fly ash | continuous | [0.0, 200.1] |
| water | continuous | [121.8, 247.0] |
| superplasticizer | continuous | [0.0, 32.2] |
| coarse aggregate | continuous | [801.0, 1145.0] |
| fine aggregate | continuous | [594.0, 992.6] |
| compressive strength | objective | maximize |

Table 4: HPLC dataset specification.

| Variable | Type | Range |
|---|---|---|
| sample_loop | continuous | [0.00, 0.08] ml |
| additional_volume | continuous | [0.00, 0.06] ml |
| tubing_volume | continuous | [0.10, 0.90] ml |
| sample_flow | continuous | [0.50, 2.50] ml/min |
| push_speed | continuous | [80.0, 150.0] Hz |
| wait_time | continuous | [1.0, 10.0] s |
| peak area | objective | maximize |

**Oracle construction protocol.** For consistency, datasets are treated as black-box objectives over their observed parameter ranges. LNP3 requires no interpolation since it fully enumerates all parameter combinations. For Cross-barrel, Concrete, and HPLC, if a complete grid is detected, $N$-dimensional multilinear interpolation is applied; otherwise, scattered data interpolation with kNN+IDW ($k = 12$, exponent $p = 2.0$, tolerance $\varepsilon = 10^{-12}$) is used.

## D ADDITIONAL EXPERIMENTAL RESULTS

### D.1 LNP3

Figures 6-8 present the scatter plots of sampled points for the three methods on the LNP3 task. Each plot is a pairplot in the normalized input space: off-diagonal panels show pairwise projections of the six variables, with points colored by optimization round (darker = earlier, lighter = later), while diagonal panels display marginal histograms. This visualization enables us to directly compare how different methods explore the search space.

Compared with **GPBO**, which scatters samples broadly across the space and exhibits large run-to-run variability, and **LLAMBO**, which provides a warmer start but still explores many uninformative regions, **LGBO under expert LLM guidance concentrates sampling much faster into promising subregions**. This behavior avoids wasting evaluations in irrelevant areas and explains why LGBO achieves both lower variance across runs and faster convergence, even though the exact optimum is unknown in real-world experiments.

### D.2 HPLC

As in the LNP3 case, Figure 13 presents the convergence traces and trajectory heatmaps on the HPLC task. The overall trend is similar: LGBO converges faster and achieves higher final performance than the baselines. The main difference is that this experiment is considerably noisier, leading to much larger fluctuations across all methods.

Figures 10-12 show the pairplots of sampled points for the three methods. Compared with **GPBO**, which scatters samples broadly and exhibits high variability, and **LLAMBO**, which provides a

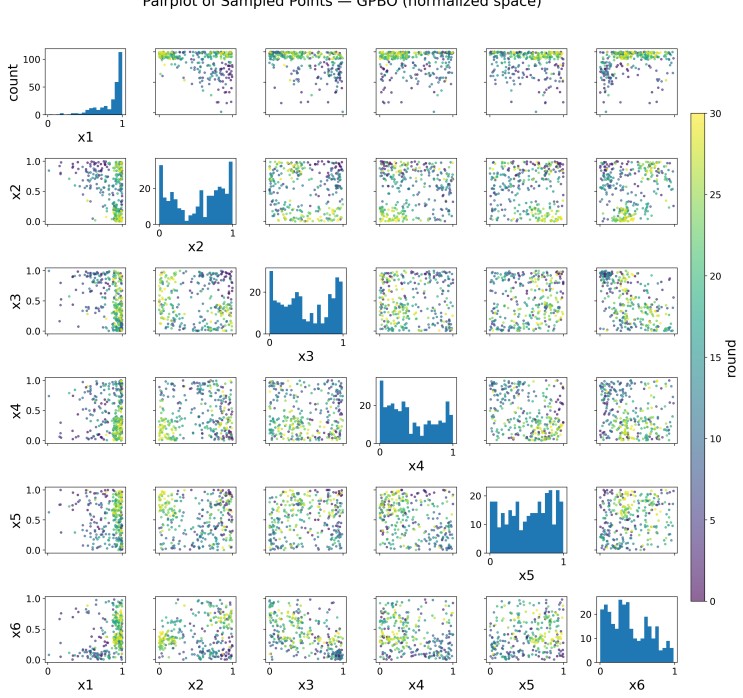

Figure 6: Pairplot of sampled points for GPBO on LNP3 (normalized space).

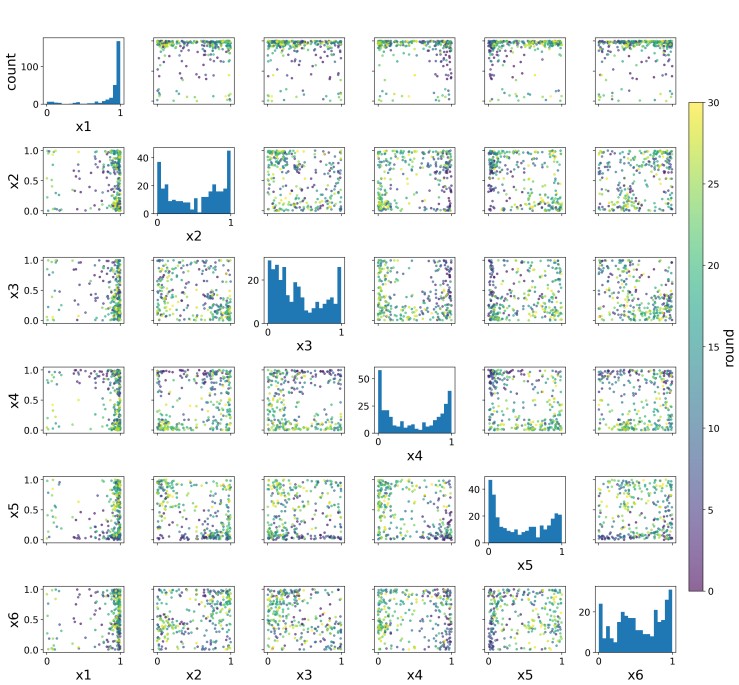

Figure 7: Pairplot of sampled points for LLAMBO on LNP3 (normalized space).

warmer start but still explores many uninformative regions, **LGBO** quickly concentrates sampling into promising subregions under expert LLM guidance. This targeted exploration avoids wasting evaluations and explains why LGBO achieves both faster convergence and lower variance across runs, even under noisy real-world conditions.

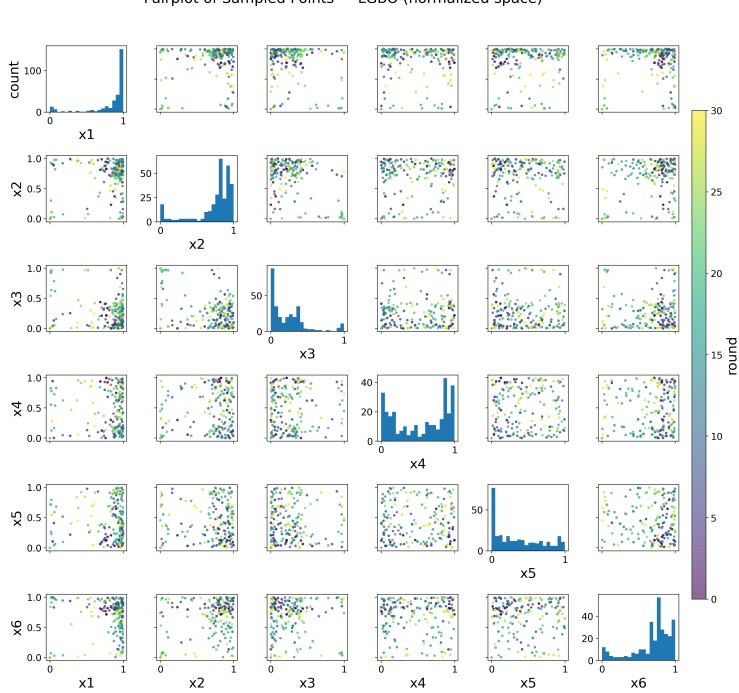

Figure 8: Pairplot of sampled points for LGBO (ours) on LNP3 (normalized space).

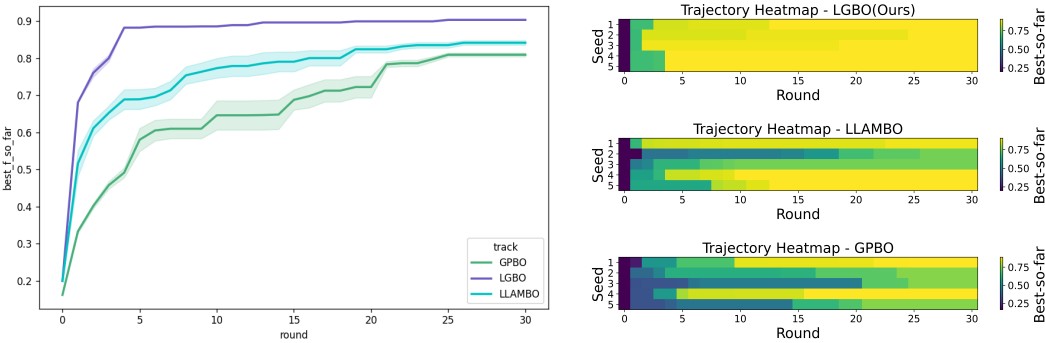

Figure 9: LNP3 results. Left: LGBO achieves faster convergence and higher final performance than GPBO and LLAMBO. Right: trajectory heatmaps show that LGBO converges more rapidly and consistently across seeds.

## D.3 CROSS BARREL

For the Cross-barrel task (a 3D-printed structural optimization with only four variables), the search space is relatively low-dimensional and thus easier for Bayesian optimization. As shown in Figure 14, even pure GPBO quickly reaches strong performance. Nevertheless, LGBO consistently achieves the best results across runs. In Figures 15-17, one interesting observation is that the scatter plots appear more dispersed in later rounds: this is because the search rapidly approaches the optimum, after which additional rounds tend to explore more broadly around the high-performing region.

## D.4 CONCRETE

For the Concrete mixture design task, in Figure 18, a widely used engineering benchmark, the LLM warm-up already lifts the initial performance to a high level. Interestingly, LGBO appears slightly

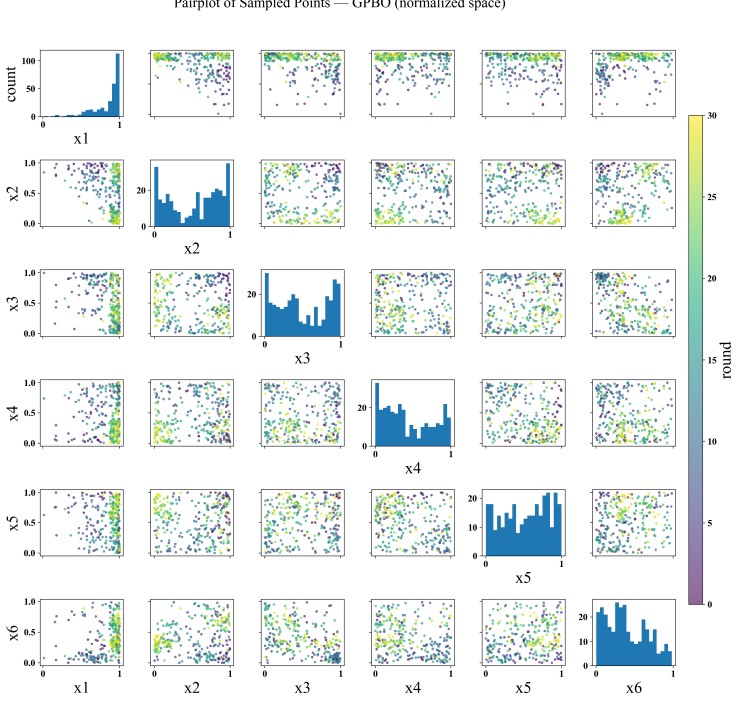

Figure 10: Pairplot of sampled points for GPBO on HPLC (normalized space).

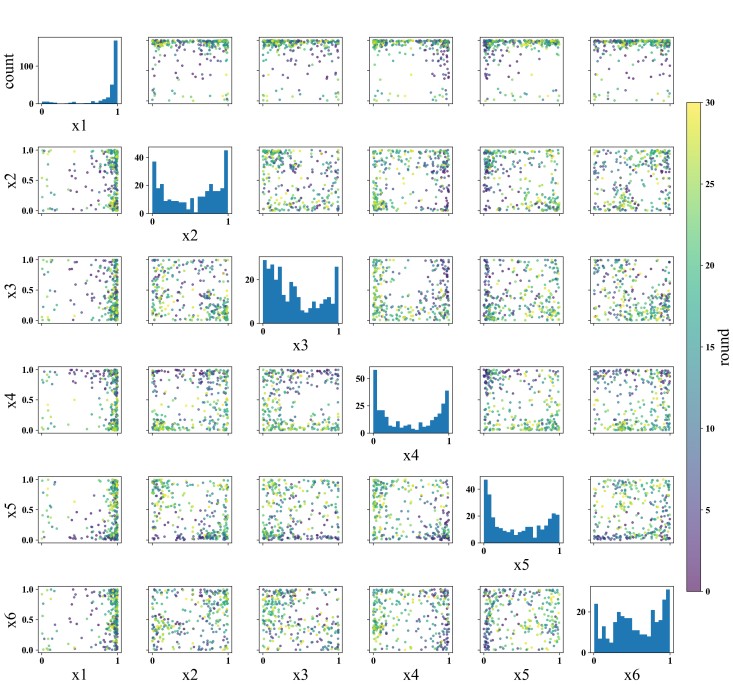

Figure 11: Pairplot of sampled points for LLAMBO on HPLC (normalized space).

weaker than LLAMBO in the very early rounds. We attribute this to the fact that LGBO, guided by the LLM, is directed toward higher-value regions rather than over-exploiting the initial local optima. As a result, although LLAMBO temporarily maintains an advantage, LGBO rapidly catches up and ultimately surpasses both baselines, achieving the highest final objective values.

Pairplot of Sampled Points — LGBO (normalized space)

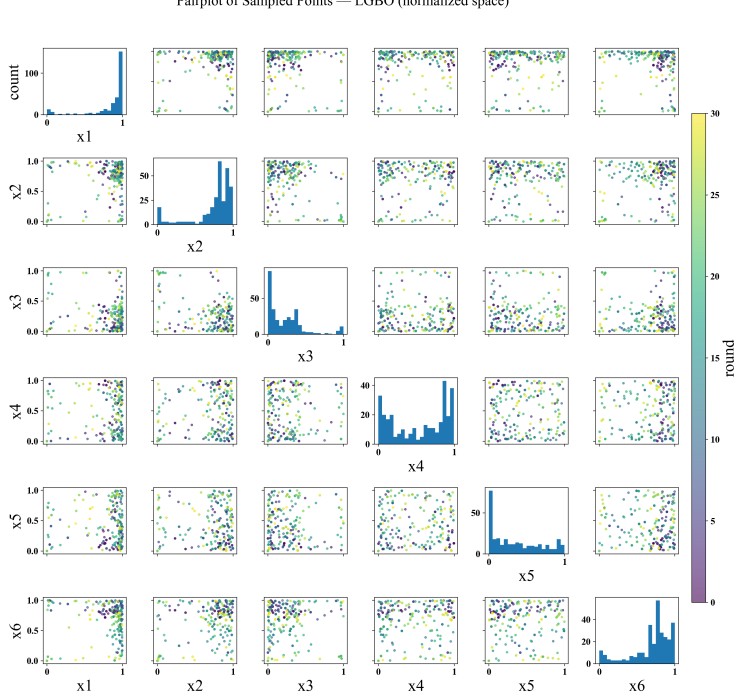

Figure 12: Pairplot of sampled points for LGBO (ours) on HPLC (normalized space).

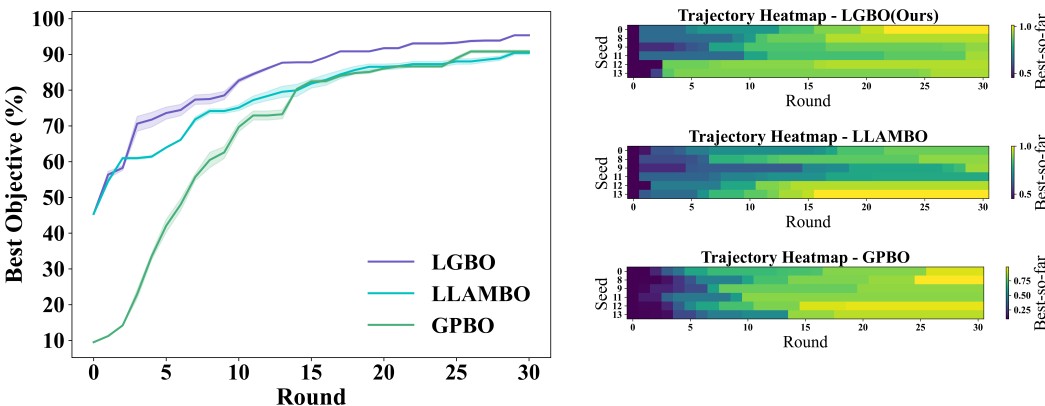

Figure 13: HPLC results. Left: mean performance traces (mean $\pm$ variance over five runs). Right: trajectory heatmaps showing convergence dynamics across seeds.

## LARGE LANGUAGE MODEL USAGE STATEMENT

In accordance with the ICLR 2026 Author Guidelines on the use of large language models, we acknowledge that large language models were used for phrasing refinement and grammar correction during manuscript preparation. All scientific ideas, algorithmic designs, and experimental results are solely the work of the authors.

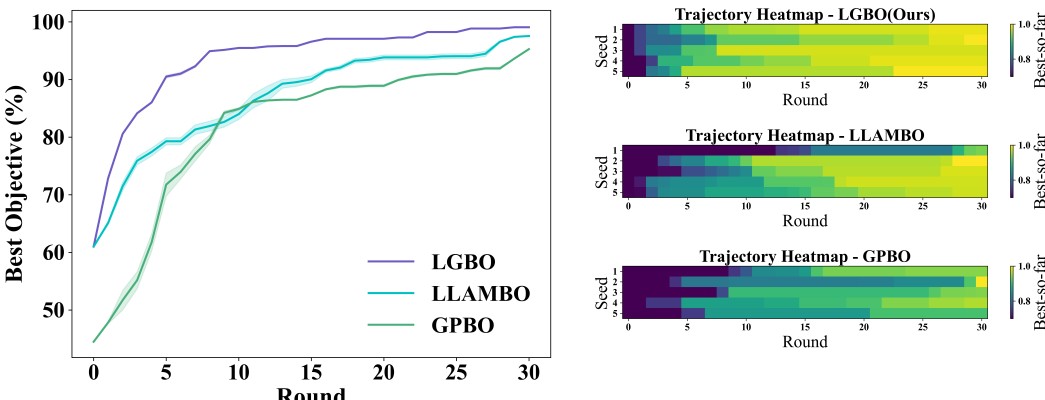

Figure 14: Cross Barrel results. Left: LGBO achieves faster convergence and higher final performance than GPBO and LLAMBO. Right: trajectory heatmaps show that LGBO converges more rapidly and consistently across seeds.

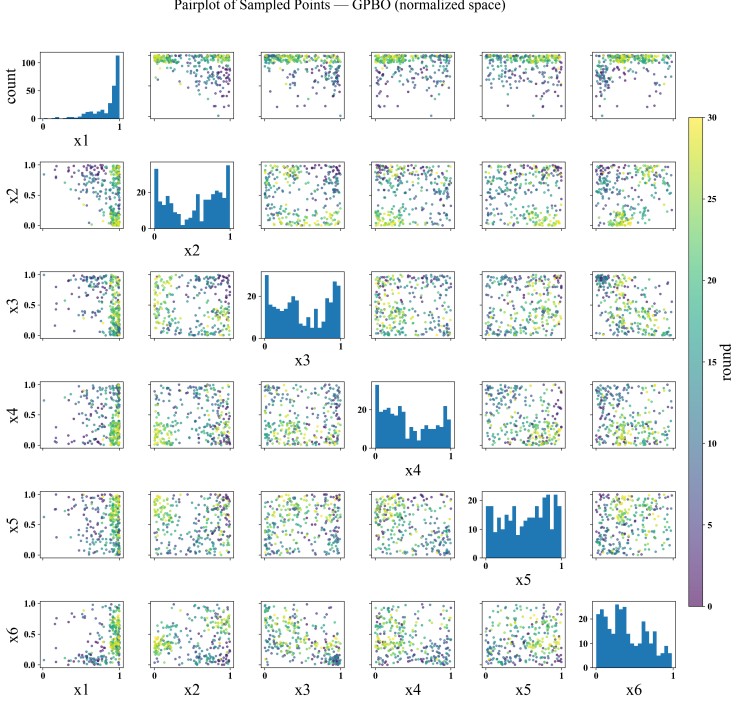

Figure 15: Pairplot of sampled points for GPBO on Cross Barrel (normalized space).

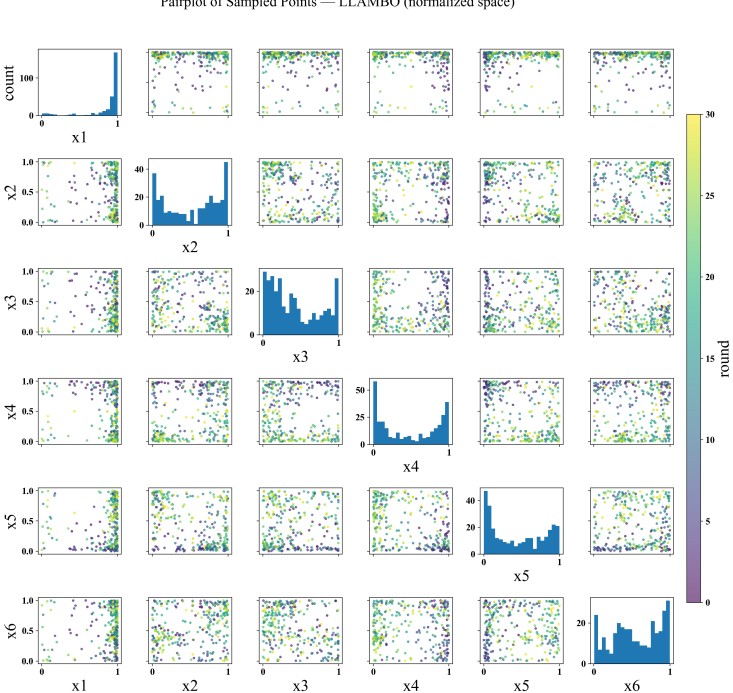

Figure 16: Pairplot of sampled points for LLAMBO on Cross Barrel (normalized space).

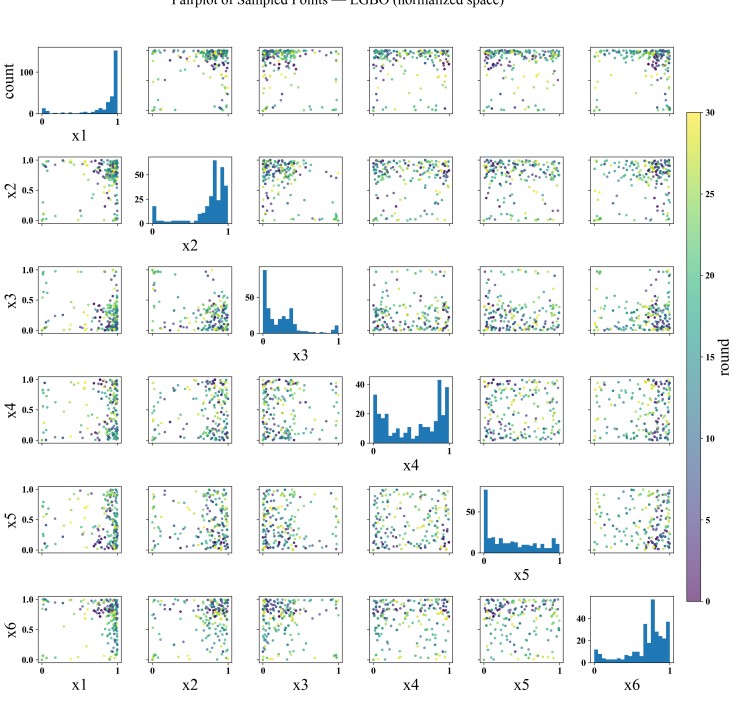

Figure 17: Pairplot of sampled points for LGBO (ours) on Cross Barrel (normalized space).

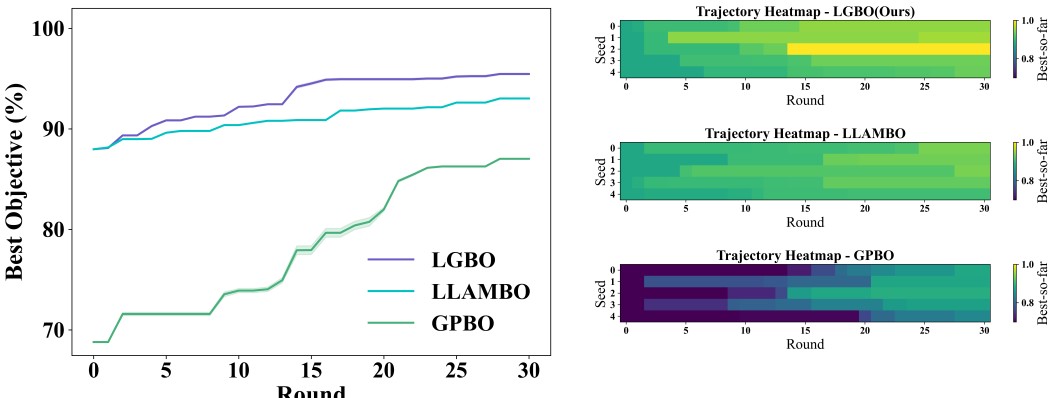

Figure 18: Concrete results. Left: LGBO achieves faster convergence and higher final performance than GPBO and LLAMBO. Right: trajectory heatmaps show that LGBO converges more rapidly and consistently across seeds.

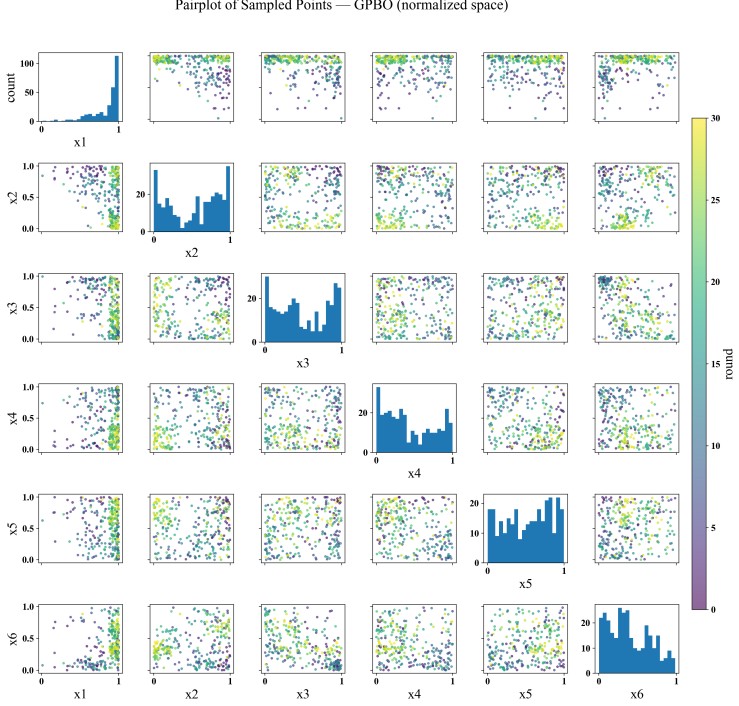

Figure 19: Pairplot of sampled points for GPBO on Concrete (normalized space).

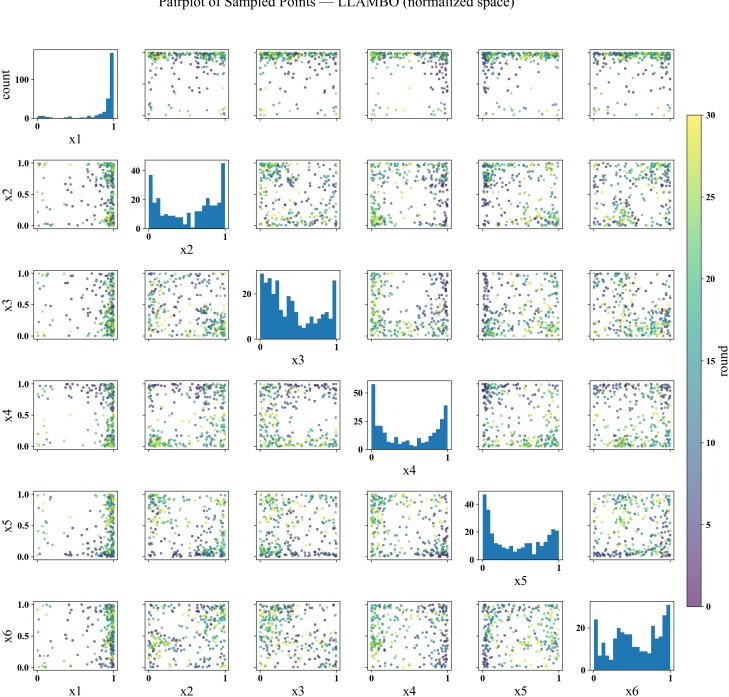

Figure 20: Pairplot of sampled points for LLAMBO on Concrete (normalized space).

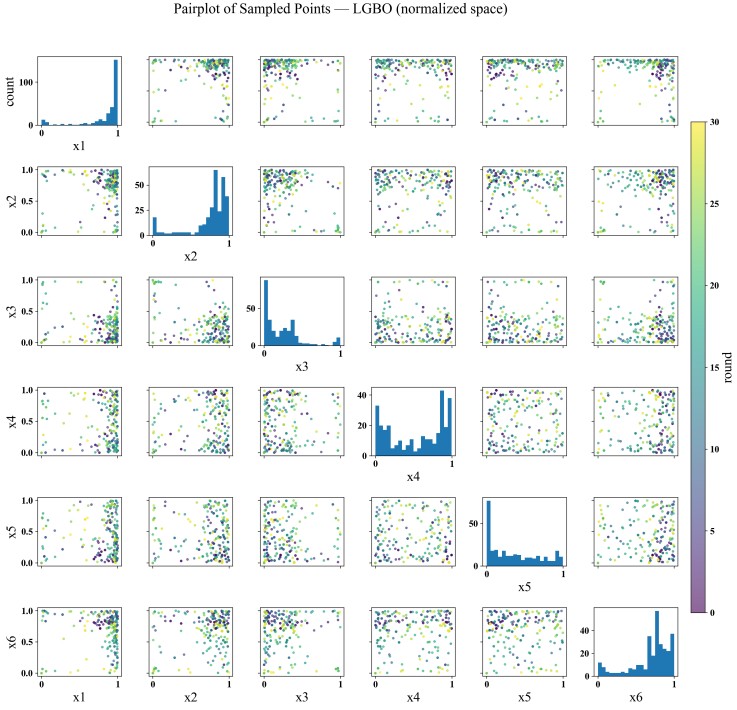

Figure 21: Pairplot of sampled points for LGBO (ours) on Concrete (normalized space).

# E    MORE EXPERIMENTS

To further evaluate the robustness and generalization capability of LGBO in scientific discovery tasks, we extend our experimental analysis with two additional baselines and a higher-dimensional benchmark. Specifically, we incorporate two recently proposed LLM-augmented Bayesian optimization methods and conduct experiments on a challenging 14-dimensional scientific dataset. And two newly LLM-BO baselines are compared with LGBO.

We conduct fair comparisons with these methods on the original benchmark datasets and further validate LGBO on a more complex 14-dimensional scientific exploration task. All extended experiments consistently demonstrate that LGBO maintains a clear performance advantage—not only in low-dimensional settings but also in high-dimensional, real-world scientific scenarios—highlighting its effectiveness, scalability, and superiority in scientific discovery applications.

## E.1    MORE DATASETS

**COF (Gantzler et al., 2023).**    The COF dataset consists of 608 covalent organic framework (COF) candidates evaluated for their performance in xenon/krypton separations. Each material is encoded as a 14-dimensional feature vector that captures its pore geometry (e.g., pore diameter, void fraction), textural properties (e.g., surface area, crystal density), and elemental composition. The optimization objective is the Xe/Kr selectivity, a central metric in adsorption-based noble-gas separations that quantifies a material's thermodynamic preference for xenon relative to krypton. Selectivity is defined as the ratio of xenon uptake to krypton uptake under identical pressure and temperature conditions, and higher values indicate stronger differential affinity toward xenon. This property is particularly important for applications such as the capture of radioactive xenon released during nuclear fuel reprocessing, off-gas treatment, and environmental monitoring, where efficient enrichment of xenon over krypton is critical.

Table 5: COF dataset specification

| Variable | Type | Range |
|---|---|---|
| pore_diameter | continuous | [3.51, 56.40] Å |
| void_fraction | continuous | [0.164, 0.928] |
| surface_area | continuous | $[1996.63, 6357.01] \, \mathrm{m^2 \, g^{-1}}$ |
| crystal_density | continuous | $[102.72, 1610.70] \, \mathrm{kg \, m^{-3}}$ |
| B | continuous | [0.000, 0.182] mol fraction |
| O | continuous | [0.000, 0.250] mol fraction |
| C | continuous | [0.325, 0.667] mol fraction |
| H | continuous | [0.000, 0.500] mol fraction |
| Si | continuous | [0.000, 0.0295] mol fraction |
| N | continuous | [0.000, 0.333] mol fraction |
| S | continuous | [0.000, 0.143] mol fraction |
| P | continuous | [0.000, 0.0667] mol fraction |
| halogens | continuous | [0.000, 0.286] mol fraction |
| metals | continuous | [0.000, 0.0238] mol fraction |
| Xe/Kr selectivity | objective | maximize |

## E.2    MORE RESULTS

We compare with two more LLM-BO algorithms below:

- **ColaLLM** (Hvarfner et al., 2024): An enhanced variant of the classical colaBO framework that integrates large language models (LLMs) to model expert preferences and employs an LLM-based warm-up strategy to generate high-quality initial candidates, thereby accelerating convergence.
- **BOPRO** (Agarwal et al., 2025): A method that leverages large language models as implicit Bayesian priors through in-context learning. BOPRO constructs prompts containing

previously observed input–output pairs and uses the LLM's contextual reasoning ability to extrapolate patterns or trends in the objective function.

- **CAKE** (Suwandi et al., 2025) A method that accelerates Bayesian optimization by designing kernel functions informed by large language models.

Across all five scientific datasets (Figure 22-26), LGBO consistently achieves the highest or near-highest objective values, particularly demonstrating strong early-stage convergence and stable final performance.

On the Cross-barrel and LNP3 tasks, LGBO rapidly surpasses other methods within the first 10 rounds, showing clear sample efficiency under limited evaluations. Although ColaLLM and LLAMBO also leverage LLM guidance, their convergence is slower and exhibits greater variance, suggesting less effective utilization of model priors.

In HPLC, LGBO steadily outperforms other approaches throughout the entire optimization process, with narrow confidence bands indicating robustness. On the COF dataset—the highest-dimensional task (d=14)—LGBO again achieves the best overall performance, highlighting its scalability and effectiveness in more challenging, high-dimensional search spaces.

The Concrete dataset appears relatively saturated across methods, yet LGBO still maintains a late-stage advantage, pointing to better exploitation capability in the final rounds.

GPBO, the only method without LLM guidance, generally lags behind the LLM-aided methods across most tasks, particularly where prior knowledge is essential or the search space is complex. While not uniformly the worst, its performance gap illustrates the tangible benefit of incorporating even lightweight model-derived preferences.

Overall, LGBO shows the most consistent, sample-efficient, and scalable optimization behavior, making it well-suited for complex scientific design settings.

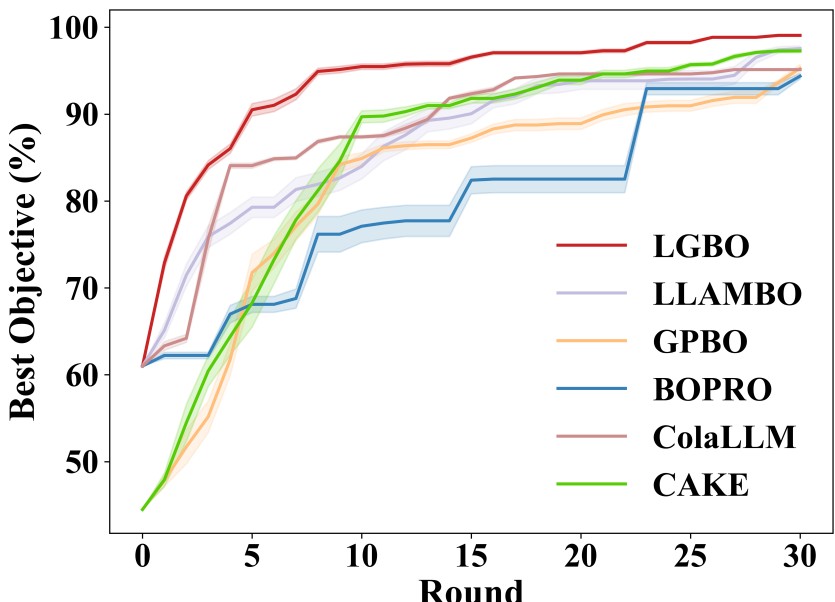

Figure 22: Performance comparison of LGBO against LLAMA, GPBO, ColaLLM and BOPRO on the Cross-barrel scientific dataset.

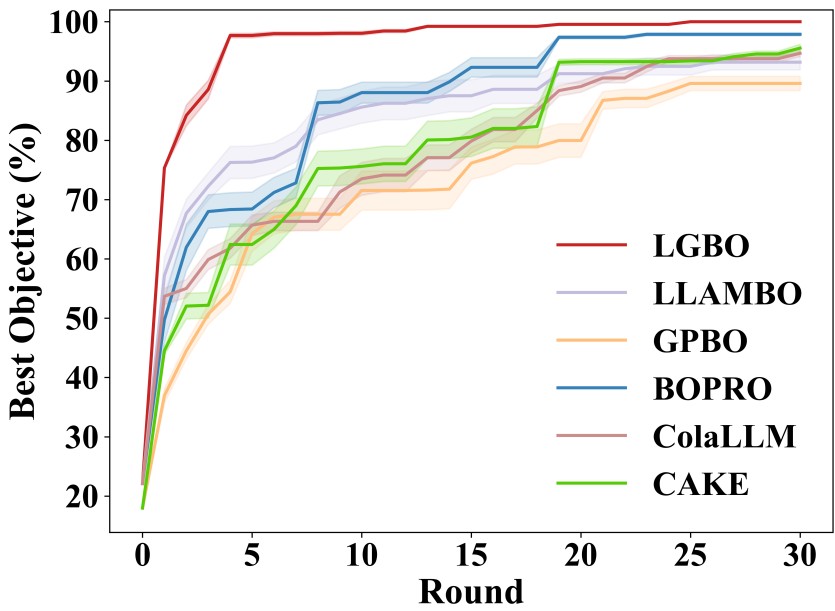

Figure 23: Performance comparison of LGBO against LLAMA, GPBO, ColaLLM and BOPRO on the LNP3 scientific dataset.

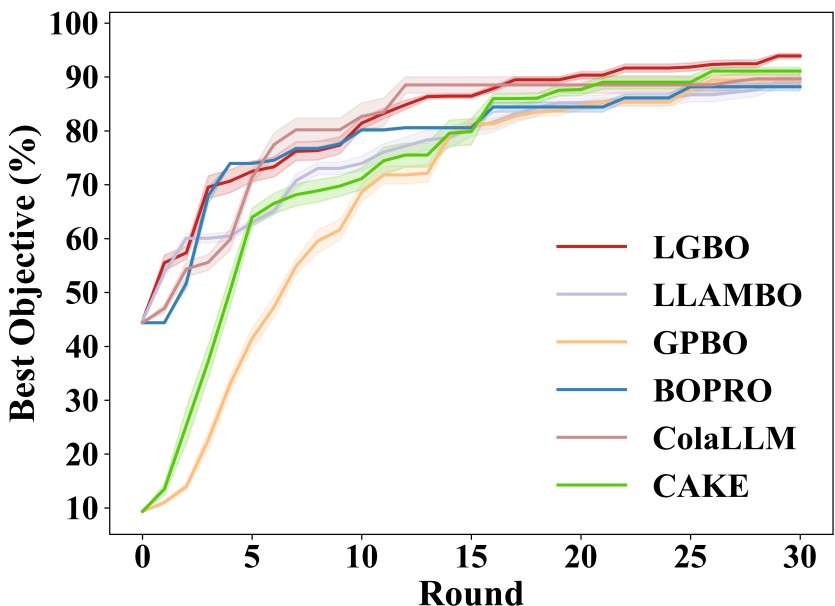

Figure 24: Performance comparison of LGBO against LLAMA, GPBO, ColaLLM and BOPRO on the HPLC scientific dataset.

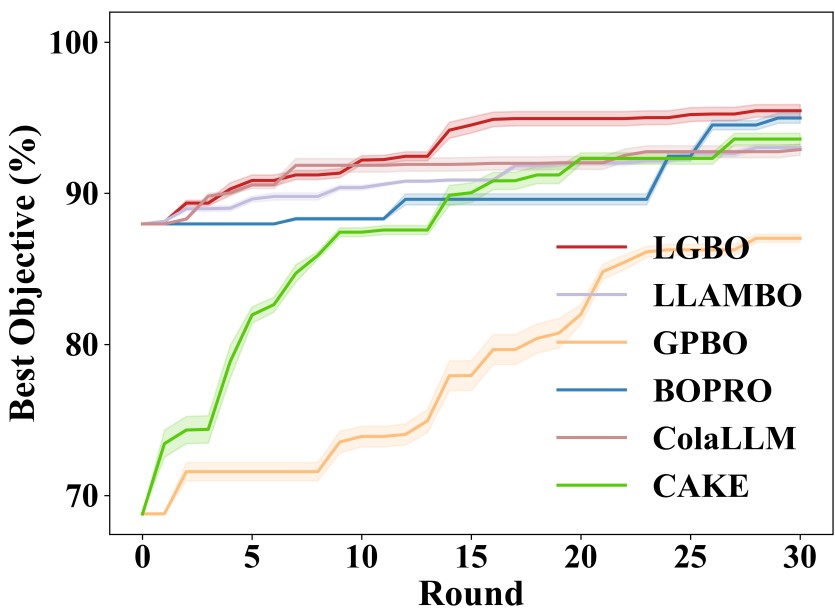

Figure 25: Performance comparison of LGBO against LLAMA, GPBO, ColaLLM and BOPRO on the Concrete scientific dataset.

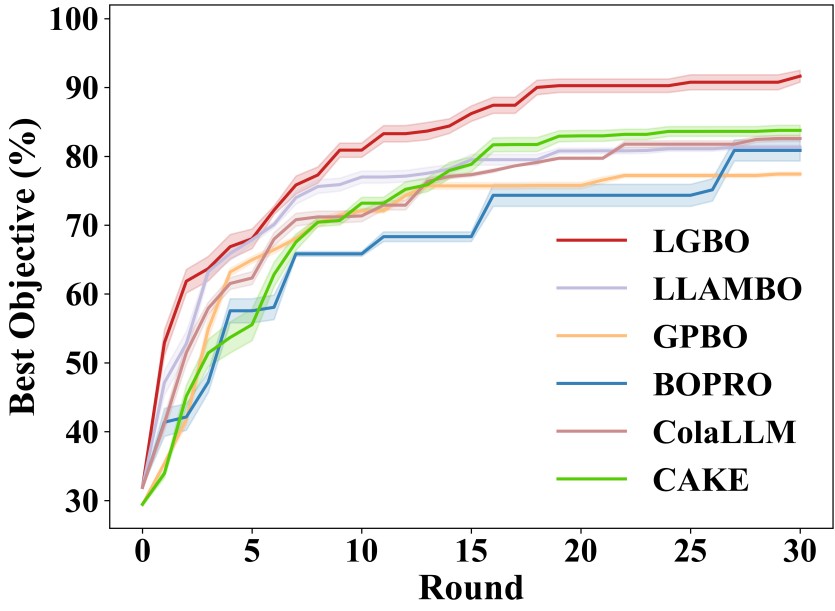

Figure 26: Performance comparison of LGBO against LLAMA, GPBO, ColaLLM and BOPRO on the COF scientific dataset.

