# OpenReview forum: "Unleashing LLMs in Bayesian Optimization: Preference-Guided Framework for Scientific Discovery"
_ICLR.cc/2026/Conference — ICLR 2026 Poster_

### Official Review · Reviewer_oNUD · 2025-10-17

**Soundness:** 2
**Presentation:** 3
**Contribution:** 3
**Rating:** 4
**Confidence:** 5

**Summary:**

This paper introduces LLM-Guided Bayesian Optimization (LGBO), a novel framework for integrating LLMs into the BO loop. The core contribution is a "region-lifted preference" mechanism, where the LLM provides continuous guidance by suggesting promising regions in the search space. This guidance is incorporated as a stable mean shift in the GP surrogate model, leaving the covariance structure intact. The authors provide theoretical regret bounds demonstrating that the method is safe in the worst case (i.e., with misaligned guidance) and can significantly accelerate convergence when the LLM's guidance is accurate. The framework is evaluated on several "dry" scientific optimization benchmarks and a "wet-lab" experiment, showing improved performance over standard GPBO and a prior LLM-augmented method, LLAMBO.

**Strengths:**

- The core idea of using an exponential lift on a discretized region, and showing its equivalence to a GP mean shift, is novel and technically sound
- Theorem 1 is useful, as it formally establishes that the framework is robust to poor/misleading LLM guidance (worst-case) and can provably accelerate convergence when guidance is informative
- The wet experiment on Fe-Cr battery electrolyte optimization is a great example to demonstrate the method's applicability to real scientific discovery problems beyond simulated benchmarks

**Weaknesses:**

- The paper motivates the work by citing the poor scalability of BO in high-dimensional settings. However, the experiments are conducted in search spaces of relatively low dimensionality (i.e., all experiments are conducted in low-dimensional settings with $d \leq 7$). Hence, the effectiveness of the proposed method in a truly high-dimensional problem is not sufficiently demonstrated
- The choice of baselines is far from adequate. The authors cite LLINBO and ReasoningBO as more systematic "LLM-in-the-loop" frameworks, yet fail to include them in the comparison. Comparisons with other related LLM-driven BO approaches, such as BOPRO [1] and CAKE [2], are also missing
- To the best of my understanding, the LLM's guidance is limited to a single point or a hyper-rectangular region. I believe this format may be too simplistic for real-world problems where promising regions could be complex (e.g., non-convex, disjoint, or like a manifold). Hence, it is not clear if this method can be applied to problems with more complex geometries

[1] D. Agarwal et al., "Searching for optimal solutions with LLMs via Bayesian optimization," ICLR, 2025.

[2] R. C. Suwandi et al., "Adaptive kernel design for Bayesian optimization is a piece of CAKE with LLMs," arXiv preprint arXiv:2509.17998, 2025.

**Questions:**

- How does LGBO handle optimization landscapes where the promising regions are have complex geometries that cannot be well-approximated by this format? Does this structural constraint limit the framework's applicability?
- How do we translate the LLM's confidence score into the guidance strength $\lambda$ and the region's properties (e.g., radius in point mode)? Since the parameter $\lambda$ is critical to the regret bounds and practical performance, the authors should provide more discussion on its selection, tuning, or sensitivity
- Based on my experience, a strong mean shift in an incorrect region without a corresponding increase in uncertainty might overly encourage the acquisition function to exploit that area. Since the proposed mechanism only shifts the surrogate's mean while leaving the covariance unchanged, could this lead to premature convergence if the LLM is confidently wrong?

---

> ### Author Response · Authors · 2025-11-20
> **Rebuttal for Reviewer oNUD (Part1)**
>
> **Dear Reviewer oNUD**,
>
> We sincerely thank you for your thoughtful and constructive feedback on our manuscript.  Below, we respond to each of your comments in detail, and we hope these revisions adequately address your concerns.
>
> ------
>
>
>
> > The paper motivates the work by citing the poor scalability of BO in high-dimensional settings. However, the experiments are conducted in search spaces of relatively low dimensionality (i.e., all experiments are conducted in low-dimensional settings with ). Hence, the effectiveness of the proposed method in a truly high-dimensional problem is not sufficiently demonstrated
>
> Thank you for raising the concern about dimensionality. We fully acknowledge the importance of scalability. In fact, we believe that LLMs are naturally well-suited for handling higher-dimensional problems, as their structured prior knowledge can provide meaningful explanations and reduce the inefficiencies associated with uninformed black-box exploration.
>
> To give an example, we have added a new 14-dimensional optimization task COF in the revised version (see Appendix E).  The COF dataset focuses on xenon/krypton separation, with each material represented by a 14-dimensional feature vector encoding pore geometry, textural properties, and elemental composition. This experiment evaluates LGBO under significantly increased dimensionality and shows that our method continues to perform stably, achieving strong gains even in this more challenging setting.
>
> In addition, although high-dimensional tasks are rare in scientific discovery, LGBO is not in conflict with mainstream high-dimensional BO methods and can be naturally combined with them. Since LGBO only modifies the posterior mean locally while preserving the GP structure, it remains fully compatible with existing approaches such as additive models [1] and trust-region methods [2], and could be integrated as a semantic guidance layer, a promising direction for future work.
>
> We hope this additional explanation helps address the reviewer’s concern regarding LGBO’s applicability beyond low-dimensional settings.
>
> [1]Kandasamy K, Schneider J, Póczos B. High dimensional Bayesian optimisation and bandits via additive models[C]//International conference on machine learning. PMLR, 2015: 295-304.
>
> [2]Eriksson D, Pearce M, Gardner J, et al. Scalable global optimization via local Bayesian optimization[J]. Advances in neural information processing systems, 2019, 32.
>
> ------
>
>
>
> > The choice of baselines is far from adequate. The authors cite LLINBO and ReasoningBO as more systematic "LLM-in-the-loop" frameworks, yet fail to include them in the comparison. Comparisons with other related LLM-driven BO approaches, such as BOPRO [1] and CAKE [2], are also missing
>
> We thank the reviewer for this insightful suggestion.
>
> To strengthen our empirical evaluation, we have extended our baseline comparisons to include several recent LLM-guided Bayesian optimization methods. Specifically, we added **BOPRO** and **CAKE**, two strong and representative approaches in the domain. Additionally, we introduced **colaLLM**, a variant of the classical **colaBO** method, where the original reliance on human expert preferences is replaced by an LLM-based expert model. This modification enables a fair comparison under the shared assumption of an LLM-in-the-loop setting.
>
> While we initially cited **LLINBO** and **ReasoningBO** as promising systematic frameworks, we were unfortunately unable to include them in our experimental comparison. These methods are currently only available as arXiv preprints and, to date, lack public implementations or sufficient technical details required for accurate reproduction. Despite considerable efforts, we could not implement them in a way that guarantees a fair or faithful comparison.
>
> The extended experiments include all previously evaluated tasks as well as challenging high-dimensional problems such as COF (d = 14). As reported in **Appendix E**, **LGBO** consistently outperforms all baselines across these tasks, demonstrating robust performance and scalability.
>
> We encourage the reviewer to refer to Appendix E in the revised manuscript for detailed results and appreciate the constructive feedback that helped us improve the rigor of our evaluation.

---

> ### Author Response · Authors · 2025-11-20
> **Rebuttal for Reviewer oNUD (Part2)**
>
> > To the best of my understanding, the LLM's guidance is limited to a single point or a hyper-rectangular region. I believe this format may be too simplistic for real-world problems where promising regions could be complex (e.g., non-convex, disjoint, or like a manifold). Hence, it is not clear if this method can be applied to problems with more complex geometries
>
> > How does LGBO handle optimization landscapes where the promising regions are have complex geometries that cannot be well-approximated by this format? Does this structural constraint limit the framework's applicability?
>
> Thank you for raising this important point regarding dimensionality.
>
> To clarify, LGBO is fundamentally a preference-guided Bayesian optimization framework: the LLM is not expected to precisely identify the true high-value region nor to approximate its full geometry.
> Instead, the LLM provides a coarse semantic preference, expressed as either a point or a simple hyper-rectangular region, that indicates where exploration might be more promising. This format provides a lightweight and interpretable guidance signal. Meanwhile, the GP surrogate and acquisition function retain full responsibility for fine-grained exploration and exploitation. The simplicity of this preference format is intentional. In our discussions with domain experts, we found that simple and interpretable region suggestions are far more useful in real scientific workflows than complex geometric shapes.
>
> Experimental scientists rarely specify non-convex or manifold-like regions; what they need is a reviewable, mechanism-consistent hint (e.g., “move toward higher pH and lower catalyst loading”), not a detailed geometric boundary. Our design fits precisely this requirement. As shown in Appendix B, each LLM-generated point or region is accompanied by natural-language reasoning, allowing experts to examine and validate the underlying rationale. This interpretability is essential in laboratory settings, where every suggested direction must be scientifically plausible and auditable.
>
> We have clarified this motivation in the revision, and we thank the reviewer for prompting us to make this aspect more explicit.
>
> ------
>
>
>
> > How do we translate the LLM's confidence score into the guidance strength and the region's properties (e.g., radius in point mode)? Since the parameter is critical to the regret bounds and practical performance, the authors should provide more discussion on its selection, tuning, or sensitivity
>
> Thank you for raising this important question.
>
> Due to space constraints, our original submission omitted this implementation detail from the main text. We have now added it back in the revised manuscript (Section 3.2, highlighted in blue).
>
> The LLM’s confidence score $c \in [0, 1]$ is mapped to the guidance strength $\lambda$ in a way that ensures the semantic lift is well-calibrated with respect to the model’s posterior uncertainty.
>
> Concretely, from Proposition 3.1, the expected effect of the regional lift is to shift the GP surrogate’s regional functional  by
> $$
> \Delta(\lambda) = \lambda a^\top \Sigma_{\mathcal{GG}} a,
> $$
> where $a$ is the region structure weight vector (uniform for region mode, locally decaying for point mode), and $\Sigma_{\mathcal{GG}}$ is the GP posterior covariance matrix on the region $\mathcal G$.
>
> To ensure that the actual magnitude of this shift reflects the LLM’s confidence level $c$, we define
> $$
> \lambda = \frac{c}{\sqrt{a^\top \Sigma_{\mathcal{GG}} a}},
> $$
> so that the **expected shift $\Delta(\lambda)$ equals $c$ standard deviations** of the regional functional. This normalization yields three benefits:
>
> - **Consistency:** The guidance strength is always scaled to reflect the LLM's confidence in a statistically meaningful way.
> - **Automatic uncertainty adaptation:** If the model is already confident in the region (low posterior variance), $\lambda$ becomes small; if the region is unexplored (high variance), $\lambda$ increases, enabling stronger influence.
> - **No manual tuning needed:** The mapping requires no extra hyperparameters or heuristics.
>
> This mechanism guarantees that the lift integrates smoothly with the model's uncertainty, **preserving theoretical soundness** (as detailed in Section 3.4) and **preventing overconfident exploitation**, even when the LLM is confidently wrong.
>
> We appreciate the reviewer’s comment, which prompted us to make this calibration strategy more explicit in the paper.

---

> ### Author Response · Authors · 2025-11-20
> **Rebuttal for Reviewer M7Dg (Part3)**
>
> > Based on my experience, a strong mean shift in an incorrect region without a corresponding increase in uncertainty might overly encourage the acquisition function to exploit that area. Since the proposed mechanism only shifts the surrogate's mean while leaving the covariance unchanged, could this lead to premature convergence if the LLM is confidently wrong?
>
> Thank you for raising this important concern.
> The mean shift introduced by LGBO is best understood as a bounded preference signal, not a hard directive that overrides the GP posterior. Even if the LLM is confidently wrong, the acquisition function operates over the same uncertainty structure as in standard BO, as the covariance is deliberately kept unchanged. This design ensures that exploitation is not artificially increased, the model is not “more certain” about the lifted region, but simply receives a modest directional bias.
>
> In fact, modifying only the mean without inflating confidence serves as a safeguard rather than a limitation. If the LLM guidance is incorrect, the acquisition function detects the mismatch between the lifted mean and observed data, prompting exploration to shift away and thereby preventing premature convergence.
>
> Our theoretical analysis formalizes this behavior. In the worst case, the regret increases only through an enlarged norm radius (bounded by $B_0 + \lambda|g|$), while the exploration guarantees of standard GP-UCB are preserved. **This means that the worst case leads to slower convergence, not convergence to an incorrect optimum**.
>
> Intuitively, LGBO introduces a “soft semantic nudge,” not an overpowering push; the uncertainty remains the true driver of exploration. We have clarified this motivation in the revision and thank the reviewer for prompting us to make this point more explicit.
>
> We sincerely appreciate your detailed and constructive review. Should you have any further questions or suggestions, we would be pleased to continue the discussion during the rebuttal period.

---

> ### Comment · Reviewer_oNUD · 2025-11-21
> **Reply to Authors' Rebuttal**
>
> I have read the rebuttal, and the authors have satisfactorily addressed my concerns. I appreciate their efforts in clarifying parts of the paper and conducting additional experiments within the stipulated time. I am happy to increase my score.

---

> > ### Author Response · Authors · 2025-11-21
> > **Reply to Reviewer oNUD**
> >
> > Thank you for taking the time to review our rebuttal and for reconsidering your evaluation. Your constructive feedback has been very helpful in improving the clarity and rigor of the paper. We sincerely appreciate your review.

---

### Official Review · Reviewer_M7Dg · 2025-10-18

**Soundness:** 2
**Presentation:** 1
**Contribution:** 3
**Rating:** 2
**Confidence:** 3

**Summary:**

This paper introduces LLM-Guided Bayesian Optimization (LGBO), a new framework that continuously integrates LLM preferences into the BO process. Unlike prior work that uses LLMs only for warm-start initialization or candidate proposal, LGBO employs a region-lifted preference mechanism that adjusts the GP surrogate mean based on LLM-specified regions of interest. The authors show that this modification maintains theoretical rigor, preserving covariance structure and provide regret bounds demonstrating that LGBO performs no worse than standard BO in the worst case and converges faster when LLM preferences align with the true objective.

Empirically, LGBO is evaluated on four dry scientific benchmarks (LNP3, Cross-barrel, Concrete, HPLC) and a new wet-lab experiment on Fe–Cr battery electrolytes. Across all tasks, LGBO outperforms both standard GP-based BO and LLMABO, showing faster convergence, higher final performance, and lower variance. Ablation studies further confirm that performance gains arise from the continous preference integration rather than initialization or random region lifting.

**Strengths:**

- Novel mechanism for incorporating LLM guidance into BO: the region-lifted preference is mathematically principled and computationally tractable. Making UCB-type bound natural to derive.
- This paper is well-motivated, with a wide range of real-world scientific discovery problems tested in the experiments.
- The methodology in this paper provides new insights about how **continuously** incorporating LLM preference could stabilize BO convergence compared to LLMABO, which integrates LLM preference indirectly, with robust ablation study.
- The framework is general and modular, compatible with standard GP surrogates and acquisition functions.

**Weaknesses:**

**The writing can be improved a lot**
- The definitions and assumptions are not stated in the main paper. The definitions of $||\cdot||$ in Line 293 and $R_T$ in Line 302 are missing, and the assumption for GP regression in the RKHS setting is not clearly stated in the main paper.

- The statement _“running GP-UCB on residual labels”_ in Line 295 is vague. Moreover, how it is equivalent to executing the proposed LGBO algorithm may be unclear to readers at first glance.

- The definition of _alignment_ being equivalent to $c > 0$ in Theorem 1 is not clearly stated and explained. Its definition appears only in the proof of Theorem 1 in Appendix A.

- In addition, there is a major typo in Theorem 1. According to the proof,
    $c = \frac{\langle f - \tau, g \rangle}{||f - \tau||\cdot||g||}$,
    while in the main text it is written as
    $c = \frac{\langle f - \mu, g \rangle}{||f - \mu||\cdot||g||}$.
    This is misleading, since $\mu$ is typically the GP mean, whereas $\tau$ is related to the lifting function.

**Theoretical result is not consistent with the proposed method.** Theorem 2 only describes the algorithm’s behavior when the mean adjustment is a fixed
    $g(\cdot) = \sum_{i=1} a_i  k(x_i, \cdot)$ through out all iterations
    whereas the proposed algorithm proposes a different $g$ through the LLM at each BO iteration. Thus, the theoretical results only partially explain the performance of LGBO.

**Lack of baseline comparison.** Only two baselines are included, and among them, only LLAMBO is an LLM-based BO method. Other LLM-integrated BO methods mentioned in the related work, such as ColaBO and LLINBO, are not included. Therefore, the statement about the instability and potential divergence of ColaBO in Line 128 is not well-supported by evidence.

**Questions:**

1. Why $a_g$ are chosen to be greater than or equal 0? ( why don't we define a general LLM-based adjustment, not just lifting potential points, but also penalizing undesirable queries by allowing negative $a_g$s?)
2. The novelty of the region-lifted preference is not emphasized until conclusion (by searching the key word "novel"). Could the authors re-confirm this is a novel idea?

---

> ### Author Response · Authors · 2025-11-20
> **Rebuttal for Reviewer M7Dg (Part1)**
>
> **Dear Reviewer M7Dg**,
>
> We sincerely thank you for your thoughtful and detailed feedback, which reflects a deep engagement with our work. We recognize that the theoretical section in the original submission lacked sufficient clarity and completeness. Your comments were instrumental in helping us identify these issues, and in response, we have substantially revised this part to improve both its rigor and readability. Below, we respond to each of your comments in detail, and we hope these revisions adequately address your concerns.
>
> > **The writing can be improved a lot**
>
> We appreciate the reviewer’s careful reading of the theoretical section. Our submission is in the *Applications to Physical Sciences* primary area, and the initial draft intentionally compressed the main-text theoretical exposition due to space limitations and the track’s emphasis on scientific applicability rather than formal optimization theory. As a consequence, several definitions were placed in the appendix.
>
> Your comments made us realize that this structure is not sufficiently friendly for readers who are interested in the theoretical aspects of LGBO. We fully agree that the main text should be self-contained. In the revised version, we have substantially expanded and reorganized the theory section:
>
> - all missing definitions and assumptions have been brought into the main text,
> - the equivalence between “running GP-UCB on residual labels’’ and the LGBO update rule is now explained explicitly,
> - the notation typo in Theorem 1 has been corrected,
> - and short intuitive explanations have been added around Theorem 2 to clarify the relationship between the fixed-lift analysis and the algorithmic setting.
>
> All relevant modifications are highlighted in blue in the updated manuscript.
>
> We respectfully invite the reviewer to take a look at the revised theory section if convenient (section 3.4). We believe the added definitions and clarifications should address the concerns raised, but we would be grateful for any further suggestions if something still appears unclear.
>
> > **Theoretical result is not consistent with the proposed method.**
>
> Thank you for raising this important concern. We acknowledge that the earlier presentation of Theorem 2 may have caused confusion regarding the connection between our theoretical analysis and the full LGBO algorithm.
>
> The theoretical analysis in the paper is **not meant to model the full adaptive LGBO procedure**, in which the LLM may output a different region at every iteration. Instead, as now stated explicitly in the revised manuscript, the theory focuses on a **frozen-lift variant**, where a single preference direction $g$ and its associated lift $\tau(x) = \lambda g(x)$ remain fixed throughout the $T$ optimization rounds.
>
> This modeling choice offers a reasonable and practically motivated abstraction. In scientific optimization tasks, domain-informed LLMs often exhibit structurally coherent behavior: once they identify features associated with high-quality regions, their suggestions tend to remain clustered, rather than fluctuate erratically across iterations. A fixed direction therefore serves as a representative surrogate for this stable class of regional preferences, while avoiding the technical complexity of analyzing a fully adaptive sequence.
>
> This abstraction is also mathematically standard and allows us to derive **cumulative regret bounds** by applying classical GP–UCB theory to the lifted objective $f' = f - \tau$. Within this setting, the analysis precisely isolates the mechanism by which LLM guidance influences optimization:
>
> - **If the LLM’s regional preference is aligned with the true objective**, the effective RKHS radius of the lifted function $f'$ becomes smaller, leading to a **strictly tighter cumulative regret bound** than standard GP–UCB.
> - **If the preference is misleading**, the radius increases by only a controlled constant amount, ensuring that the **worst-case cumulative regret matches that of GP–UCB**.
>
> Although the frozen-lift abstraction does not fully reproduce the adaptive behavior of LGBO, it enables a sound and interpretable analysis that captures both the **accelerated convergence** when the guidance aligns with the objective and the **robustness** when it does not.
>
> We have revised Section 3.4 (highlighted in blue) to clearly state this modeling assumption and to eliminate the ambiguity present in the original version.

---

> > ### Comment · Reviewer_M7Dg · 2025-11-25
> > **response**
> >
> > I appreciate the authors' effort to modify the paper and improve the presentation. I would like to increase my score because 1) My major concern about the presentation seems to be addressed 2) the authors confirmed that their idea is novel, then I acknowledge this contribution.

---

> > > ### Author Response · Authors · 2025-11-27
> > >
> > > Thank you for taking the time to review our rebuttal and for reconsidering your evaluation. Your feedback on the rigor of the theoretical section has been immensely helpful in improving the clarity of our paper. We sincerely appreciate your review.

---

> ### Author Response · Authors · 2025-11-20
> **Rebuttal for Reviewer M7Dg (Part2)**
>
> > Why are chosen to be greater than or equal 0? ( why don't we define a general LLM-based adjustment, not just lifting potential points, but also penalizing undesirable queries by allowing negative s?)
>
> This design choice is rooted in the scientific discovery setting. In real physical/chemical optimization (including our Fe–Cr electrolyte experiment), clearly meaningless or infeasible regions known to be infeasible or irrelevant (e.g., reagent concentrations too low to produce a reaction, unsafe temperatures) are typically excluded a priori during search space definition. The LLM therefore acts as a provider of positive semantic cues The LLM therefore serves to provide positive semantic cues, for example, identifying promising regions to explore further, rather than penalizing regions already excluded by expert knowledge. Allowing negative lifts is algorithmically feasible, but it does not align with how scientists typically define feasible design spaces.
>
> > The novelty of the region-lifted preference is not emphasized until conclusion (by searching the key word "novel"). Could the authors re-confirm this is a novel idea?
>
> We respectfully confirm the novelty of our approach.
>
> Existing LLM-BO methods (LLAMBO, ReasoningBO, BOPRO) use LLMs for warm-starts or candidate proposals that are then filtered by the acquisition function. They do **not** continuously shape the GP surrogate mean via region-level preferences while preserving the covariance structure. Our framework provides such a mechanism through region-lifted mean shifts at every iteration.
>
> We again thank the reviewer for the helpful comments. The revised version incorporates the requested clarifications in the theory, corrected notation, and additional discussion and baselines where appropriate (all highlighted in blue). We hope that these changes address the concerns and make the paper more accessible to both theory-oriented and application-oriented readers.

---

### Official Review · Reviewer_W7ee · 2025-10-31

**Soundness:** 3
**Presentation:** 3
**Contribution:** 2
**Rating:** 8
**Confidence:** 3

**Summary:**

This paper introduces LLM-Guided Bayesian Optimization (LGBO), a framework that integrates priors from large language models (LLMs) into Bayesian Optimization (BO) for scientific experimentation. Unlike previous approaches that use LLMs only for the initialization of BO exploration or candidate generation, LGBO continuously incorporates LLM-derived region-lifted preferences into every iteration of the optimization loop. The authors show that this mechanism effectively shifts the Gaussian process (GP) surrogate’s mean without affecting its covariance, allowing semantic LLM guidance while retaining BO’s statistical guarantees. Theoretical results prove that LGBO is not worse than standard BO in the worst case and can achieve faster convergence when LLM guidance aligns with the true objective. Experiments on four “dry” scientific benchmarks (LNP3, Cross-barrel, Concrete, HPLC) and one “wet-lab” Fe–Cr battery optimization demonstrate consistent acceleration and improved stability over GPBO and LLAMBO baselines.

**Strengths:**

- Novel integration: Proposes a principled and theoretically grounded method to embed LLM preferences directly into the BO surrogate, moving beyond heuristic or warm-start use of LLMs.

- Theoretical formulation: Provides formal regret bounds showing bounded degradation under misalignment.

- Evaluation: Including both simulation and real-world experiments, showing convincing empirical improvements across diverse domains.

- Reproducibility: Clearly describes prompt templates, datasets, and experimental protocols; theoretical and implementation details are provided.

- Scientific relevance: Demonstrates the potential for LLMs to accelerate experimental optimization in time-limited domains.

- Stable framework design: The region-lifted preference mechanism elegantly maintains GP structure and stability, avoiding instability issues typical of preference-based methods.

**Weaknesses:**

- Evaluation baselines: Evaluation omits other preference-based or human-in-the-loop BO methods (e.g., ColaBO, Preferential BO with human experts), which could contextualize LGBO’s relative contribution.

- Dependence on prompt engineering: Performance and robustness may depend strongly on prompt quality and LLM capabilities, but sensitivity analyses on prompt design are limited.

- Scalability questions: Experiments focus on low- to medium-dimensional tasks (≤6 variables); it remains unclear how LGBO scales to high-dimensional or multi-objective settings.

- Computational cost: Continuous LLM querying at each iteration may incur significant computational or latency overhead; this is not quantified or mentioned in the paper.

- Interpretability of LLM Guidance: While the framework embeds preferences mathematically, the semantic validity or interpretability of the generated regions is not thoroughly analyzed.

- Limited real-world task diversity: Only one wet-lab experiment is included; broader real-world validations would strengthen generality claims. It is also not clear why these particular tasks were selected for the evaluation.

**Questions:**

- What is really the motivation for choosing preference optimisation for this setting? It is not immediately obvious to me. Better motivation as context for the approach would be helpful.

- The region-lifted preference is formalized as an exponential mean shift. Could similar effects be achieved using other functional forms (e.g., linear or kernel-weighted lifts), and what motivated the specific exponential choice?

- Since LGBO queries the LLM at every iteration, what is the computational or latency overhead compared to traditional BO? Could lightweight surrogates or cached reasoning traces mitigate this cost?

- How sensitive are the results to the exact prompt design or LLM choice? While Appendix B provides structured prompts, have you quantified performance variance across alternative phrasing or reasoning styles?

- Have you analyzed the semantic quality of the LLM-suggested regions—do they align with known scientific heuristics or physical laws?

---

> ### Author Response · Authors · 2025-11-20
> **Rebuttal for Reviewer W7ee (Part1)**
>
> **Dear Reviewer W7ee,**
>
> We sincerely thank you for your thoughtful and insightful review of our work. Below, we provide a point-by-point response, structured into five integrated aspects that collectively address all of your comments.
>
> ------
>
> #### **Aspect 1: Motivation and Modeling Choices**
>
> > What is really the motivation for choosing preference optimisation for this setting? It is not immediately obvious to me. Better motivation as context for the approach would be helpful.
>
> > The region-lifted preference is formalized as an exponential mean shift. Could similar effects be achieved using other functional forms (e.g., linear or kernel-weighted lifts), and what motivated the specific exponential choice?
>
> Thank you for these insightful questions. The preference-based formulation is motivated by the nature of scientific optimization tasks, where LLMs are typically unable to provide precise optima but can offer coarse, interpretable regional suggestions. These align more naturally with preference judgments over subsets of the search space. Formulating the guidance as a regional preference enables seamless integration into the BO framework while preserving robustness and convergence guarantees.
>
> In our prompt design (Appendix B), the LLM is explicitly asked to justify *why* a region is promising. This added layer of interpretability is especially valuable in scientific discovery, where understanding the rationale behind a suggestion is essential for hypothesis generation and experimental planning.
>
> Regarding the functional form, LGBO does not rely strictly on the exponential structure. As discussed in Section 3.2, the framework accommodates general preference functionals. We adopt the exponential form in this implementation because it provides smoothness, numerical stability, and analytical tractability—particularly useful for deriving regret bounds (Proposition 1). However, other lifting schemes could be explored in future work.
>
> ------
>
> #### **Aspect 2: Computational Cost and Task Scope**
>
> > Since LGBO queries the LLM at every iteration, what is the computational or latency overhead compared to traditional BO? Could lightweight surrogates or cached reasoning traces mitigate this cost?
>
> > Computational cost: Continuous LLM querying at each iteration may incur significant computational or latency overhead; this is not quantified or mentioned in the paper.
>
> > Limited real-world task diversity: Only one wet-lab experiment is included; broader real-world validations would strengthen generality claims. It is also not clear why these particular tasks were selected for the evaluation.
>
> We appreciate these important observations. While LGBO introduces an LLM query per BO iteration, in scientific optimization scenarios the cost of such queries is negligible compared to that of running real experiments. For example, in our Fe–Cr electrolyte task, each iteration corresponds to a lab experiment involving reagents, instrumentation, and time-intensive procedures. Thus, the additional LLM call (typically under 10 seconds) adds virtually no overhead in practice. That said, cached reasoning or lightweight surrogates could be incorporated in future extensions.
>
> As for task selection, we combine standard dry benchmarks with a real-world wet-lab optimization task that is both scientifically meaningful and methodologically challenging. The Fe–Cr electrolyte experiment represents an active research area where LLMs have no domain-specific training exposure, making it a strong test of generalization ability. Moreover, the improved candidates discovered through LGBO have independent scientific value and may inform future electrolyte designs.
>
> We also agree that expanding to more real-world wet-lab experiments is an important direction. While such experiments are inherently time- and resource-intensive, they offer a realistic and high-impact setting to evaluate LLM-guided optimization. We are actively working with domain collaborators to apply LGBO to additional wet-lab scenarios and plan to pursue these efforts beyond the current submission.

---

> ### Author Response · Authors · 2025-11-20
> **Rebuttal for Reviewer W7ee (Part2)**
>
> #### **Aspect 3: Prompt Robustness and Dependence on LLM Design**
>
> > How sensitive are the results to the exact prompt design or LLM choice? While Appendix B provides structured prompts, have you quantified performance variance across alternative phrasing or reasoning styles?
>
> > Dependence on prompt engineering: Performance and robustness may depend strongly on prompt quality and LLM capabilities, but sensitivity analyses on prompt design are limited.
>
> Thank you for highlighting this important consideration. Indeed, prompt design and LLM capability are known sources of variance in LLM-in-the-loop systems. Our framework addresses this challenge by imposing a structured prompt template (Appendix B) that (i) requires a concrete region proposal, and (ii) demands a scientific rationale and confidence estimate. This structure restricts open-ended generation, improving consistency and interpretability.
>
> In addition, based on our observations, current LLMs—having been exposed to a wide range of paraphrased expressions during pretraining—tend to produce relatively stable outputs in response to commonly used synonymous prompts. This further reduces the system’s sensitivity to minor wording variations in practical settings.
>
> Moreover, LGBO incorporates LLM outputs in a *controlled* manner, via a calibrated mean shift that retains the GP’s uncertainty structure. This means the acquisition function continues to guide exploration in a principled way, even if the LLM’s guidance varies slightly due to prompt or model differences. The BO loop remains stable and convergent by construction, and future versions of LGBO could incorporate automatic prompt optimization or ensemble LLM prompting to further mitigate variability.
>
> ------
>
> #### **Aspect 4: Semantic Interpretation of LLM-Provided Regions**
>
> > Interpretability of LLM Guidance: While the framework embeds preferences mathematically, the semantic validity or interpretability of the generated regions is not thoroughly analyzed.
>
> > Have you analyzed the semantic quality of the LLM-suggested regions—do they align with known scientific heuristics or physical laws?
>
> Yes, this is a central goal of our work. As shown in Appendix B, each LLM-guided iteration involves not only a region proposal but also an accompanying explanation grounded in scientific principles. This promotes transparency and interpretability, enabling human experts to audit the LLM’s reasoning and judge whether the suggestion is plausible before committing experimental resources. Our case study demonstrates that the LLM-suggested regions often align with known heuristics in electrochemistry and have led to successful discovery of improved formulations.
>
> ------
>
> #### **Aspect 5: Experiments**
>
> > Scalability questions: Experiments focus on low- to medium-dimensional tasks (≤6 variables); it remains unclear how LGBO scales to high-dimensional or multi-objective settings.
>
> >  Evaluation baselines: Evaluation omits other preference-based or human-in-the-loop BO methods (e.g., ColaBO, Preferential BO with human experts), which could contextualize LGBO’s relative contribution.
>
> Thank you for these thoughtful points. In the revised version, we have added a new **14-dimensional** optimization benchmark to assess LGBO’s scalability. This demonstrates that LGBO remains stable and effective even in substantially higher-dimensional settings.
>
> Regarding comparative evaluation: as you noted, most preference-based BO methods (e.g., ColaBO) are human-in-the-loop and do not involve language models. To enable a fair comparison, we introduce a modified version—**ColaLLM**—which adapts the ColaBO structure to accept LLM preferences. This adaptation is intended solely for ablation and benchmarking; it is not an existing method in the literature.
>
> All new experiments, setups, and results are included in the Appendix E in the revised version. We would be grateful if you could review these additions.
>
> ------
>
> **We thank you again for the thorough and insightful review.  If you have any further questions, we are willing to discuss them with you at any time during the rebuttal period.

---

### Official Review · Reviewer_935h · 2025-10-31

**Soundness:** 3
**Presentation:** 3
**Contribution:** 3
**Rating:** 6
**Confidence:** 4

**Summary:**

This paper proposes a way to integrate prior information in large language models into Bayesian optimization.
At its core, the method is based on a simple idea.
It prompts the language model to generate a promising region in the search space.
Then, the proposed method updates the GP mean function so that the mean function has higher values in the promising region.
Experiments on benchmark test functions across several areas show promising results against conventional BO.

**Strengths:**

1. This paper works in the intersection of Bayesian optimization and large language models, and shows how to effective utilize the prior information stored in LLMs.
Those prior information is often hard to encode into conventional Bayesian optimization by kernel designs.

1. Using the lifting functional to encode the prior information from the language model makes sense to me.
Though I have not checked the math, the part that the functional in the exponential form turns out to shift the mean function makes sense to me, as it is similar to exponential tilting for Gaussian distributions.

**Weaknesses:**

1. The interpretation of Theorem 1 seems to be over claimed.
The correct interpretation should be weaker than what's claimed in the paper.
The theorem assumes the lift is given in advanced, and **does not change** during BO.
However, the method proposed in this paper actually utilizes language models interactively in that the lifted region and or points get updated in each iteration of BO.
Thus, if the language model's prediction is bad, the regret bound could be unbounded.

1. Many important technical details are missing.
    - How to set the guidance strength parameters \\(a_g\\)?
    Do you estimate them during GP model fitting with maximum likelihood, or do you set them to fixed numbers?
    - How the discretization \\(x_g\\) is chosen in a lifted region?
    - How does the confidence scores generated by LLMs affect the lifting functional?

**Questions:**

1. What's the exact definition of misaligned lift in Theorem 1? Isn't the misaligned case the same as \\(c = 0\\), i.e., small cosine similarity?

1. Line 293: Is the norm in theorem 1 the RKHS norm?
If so, it would be better to make it explicit.

1. Line 162: "Intuitively, \\(p(\cdot)\\) here denotes a probability distribution...".
Shouldn't it be \\(\rho\\)?

---

> ### Author Response · Authors · 2025-11-20
> **Rebuttal for Reviewer 935h (Part1)**
>
> **Dear Reviewer** **935h**,
>
> We sincerely thank you for your support of this work. Below, we respond to each of your comments in detail, and we hope these revisions adequately address your comments.
>
> > The interpretation of Theorem 1 seems to be over claimed. The correct interpretation should be weaker than what's claimed in the paper. The theorem assumes the lift is given in advanced, and **does not change** during BO. However, the method proposed in this paper actually utilizes language models interactively in that the lifted region and or points get updated in each iteration of BO. Thus, if the language model's prediction is bad, the regret bound could be unbounded.
>
> Thank you for highlighting this important point. We acknowledge that the original phrasing may have inadvertently suggested that the regret guarantee extends to the fully adaptive LGBO setting, where the LLM proposes a new region at each iteration.
>
> In the revised manuscript (Section 3.4), we now explicitly clarify that Theorem 1 analyzes a simplified **frozen-lift** setting, in which the LLM-induced preference direction $g$ is fixed before the start of optimization, and the corresponding lift $\tau(x) = \lambda g(x)$ remains unchanged throughout the process. This is the precise setting covered by our theoretical analysis, and we are grateful to the reviewer for prompting us to make this distinction more transparent.
>
> Regarding the possibility of non-convergence or unbounded regret: we agree that, in principle, if the LLM were to produce highly confident but poorly aligned guidance at every iteration, any method relying on such signals could accumulate significant regret. However, in the scientific optimization domains we target, such erratic behavior is highly atypical. Domain-informed LLMs—when prompted with structured context—tend to generate semantically coherent suggestions that remain localized within consistent regions of the parameter space across iterations, rather than fluctuating unpredictably.
>
> Furthermore, in real-world laboratory workflows, proposed regions are reviewed by human experts before initiating expensive physical experiments. As detailed in Appendix B, our prompt design explicitly requests the LLM to cautiously justify its recommendations and to provide a confidence estimate, thereby enabling early detection and filtering of implausible or unstable guidance.
>
> These practical safeguards help mitigate the risks associated with pathological behavior and provide an additional layer of robustness beyond what is covered by the theoretical model.
>
> > Many important technical details are missing.
> >
> > - How to set the guidance strength parameters ? Do you estimate them during GP model fitting with maximum likelihood, or do you set them to fixed numbers?
> > - How the discretization is chosen in a lifted region?
> > - How does the confidence scores generated by LLMs affect the lifting functional?
>
> We appreciate the opportunity to clarify these points. The revised version now explicitly details the full lifting interface in Section 3.2 (highlighted in blue). Below, we address each component of the reviewer’s question:
>
> **(1) How the weight vector $a$ is constructed.**
>  In **region-lift mode**, the LLM-suggested region is a hyper-rectangle. The regional functional averages over this region, and the corresponding weight vector $a$ is a vector of ones over the grid points within the region.
>  In **point-lift mode**, we use a smooth localized functional: $a$ is defined by a Gaussian kernel centered at the LLM-suggested point. This yields a soft weighting around the point, enabling the lift to influence a neighborhood rather than a single location, which improves robustness and compatibility with continuous input spaces.
>
> **(2) Why the region is discretized.**
>  The discretization is inherited from the GP implementation rather than independently introduced for lifting. Specifically, our BO system employs Matheron-rule pathwise sampling based on a fixed low-discrepancy grid (e.g., Sobol sequence). When the LLM suggests a region, we intersect it with this global grid and define the weight vector $a$ over the resulting subset. This design ensures that the lifted region aligns exactly with the GP’s internal discretization, avoiding additional sampling artifacts and keeping the integration seamless within the BO pipeline.

---

> ### Author Response · Authors · 2025-11-20
> **Rebuttal for Reviewer 935h (Part2)**
>
> **(3) How the LLM confidence score determines the lift strength.**
> Due to space constraints, our original submission omitted this implementation detail. We have now added it back in the revised version (Section 3.2, highlighted in blue).
>
> The LLM’s confidence score $c \in [0, 1]$ is mapped to the guidance strength $\lambda$ in a way that ensures the semantic lift is well-calibrated with respect to the GP model’s posterior uncertainty.
>
> From Proposition 3.1, the expected effect of the lift is to shift the regional functional by
> $$
> \Delta(\lambda) = \lambda\ a^\top \Sigma_{\mathcal{GG}} a,
> $$
> where $a$ is the structure weight vector (uniform for region mode, locally decaying for point mode), and $\Sigma_{\mathcal{GG}}$ is the GP posterior covariance over the region $\mathcal G$.
>
> To ensure that this shift matches the LLM’s semantic confidence level $c$, we define
> $$
> \lambda = \frac{c}{\sqrt{a^\top \Sigma_{\mathcal{GG}} a}}.
> $$
> This calibration guarantees that the induced shift $\Delta(\lambda)$ equals **exactly $c$ standard deviations** under the model’s current belief.
>
> This construction offers several benefits:
>
> - **Statistical alignment:** The strength of the lift consistently reflects the LLM's confidence in probabilistic terms.
> - **Automatic adaptation to model uncertainty:** If the model is already confident in the suggested region (low variance), $\lambda$ becomes small; if the region is uncertain, a larger $\lambda$ ensures the same scale of shift.
> - **No manual tuning or heuristics:** The mapping requires no additional hyperparameters and remains fully analytical.
>
> This strategy integrates the LLM’s soft semantic prior with the model’s current belief in a principled way, supporting both robustness and theoretical soundness.
>
> > What's the exact definition of misaligned lift in Theorem 1? Isn't the misaligned case the same as , i.e., small cosine similarity?
>
> Thank you for the excellent question. **Yes, your understanding is correct**: the “misaligned” case in Theorem 1 indeed corresponds to **small or negative cosine similarity** between the true objective and the LLM-induced preference function.
>
> To clarify this more formally, we have added a dedicated paragraph in Section 3.4 of the revised manuscript (titled *“Alignment coefficient”*) defining the exact notion of alignment used in the theorem.
>
> We introduce the **alignment coefficient**:
> $$
> c \;:=\;
> \frac{\langle f-\tau,\, g\rangle_{\mathcal{H}_k}}
>      {\|f-\tau\|_{\mathcal{H}_k}\,\|g\|_{\mathcal{H}_k}}
>      \;\in [-1,1],
> $$
> which measures the cosine similarity between the centered objective function $f - \tau$ and the LLM’s guidance direction $g$ in the RKHS $\mathcal{H}_k$.
>
> - When $c \approx 1$, the lift is **well-aligned**, meaning the guidance pushes optimization in a direction consistent with improving values of $f$.
> - When $c \approx 0$ or negative, the lift is **misaligned**, offering little or potentially misleading information.
>
> Thus, the regret bound in Theorem 1 explicitly depends on this alignment coefficient: **a misaligned lift corresponds to small or negative $c$**, which increases the effective RKHS norm and hence loosens the bound. This formalization enables a clean theoretical separation between helpful and harmful guidance.
>
> > Line 293: Is the norm in theorem 1 the RKHS norm? If so, it would be better to make it explicit.
>
> Thank you for this helpful comment. You are absolutely correct, the norm in Theorem 1 refers to the RKHS norm induced by the kernel $k$. To avoid any ambiguity, we have made this explicit throughout Section 3.4 in the revised manuscript.
>
> > 1. Line 162: "Intuitively, p() here denotes a probability distribution...". Shouldn't it be $\rho$?
>
> Thank you for pointing this out. The original sentence was intended to introduce the probability distribution $p$ prior to defining the adjusted posterior $p(f \mid D_t, \rho)$, but we agree the phrasing was unclear and could be misread as conflating $p(\cdot)$ with the preference functional $\rho(f)$. We have removed this sentence in the revised version to eliminate the ambiguity.
>
> We thank you again for the thorough and insightful review.  If you have any further questions, we are willing to discuss them with you at any time during the rebuttal period.

---

> > ### Comment · Reviewer_935h · 2025-11-25
> >
> > Thanks for the response. A clarification question. Does the confidence score $c \in [0, 1]$ corresponds to the logit after the softmax layer in LLMs?

---

> > > ### Author Response · Authors · 2025-11-27
> > >
> > > Thank you for the thoughtful question. To clarify, the confidence score $c \in [0,1]$ used in our method is not directly derived from the softmax logits of the LLM. As defined in Section 3.2 and shown in the prompt templates in Appendix B, the LLM is instructed to output one of two formats:
> > >
> > > 1. **Point mode**: `[point, [x1, x2, ..., xd], ccc]`
> > > 2. **Region mode**: `[region, [[lb1, ..., lbd], [ub1, ..., ubd]], ccc]`
> > >
> > > Here, `ccc` is a scalar confidence score in the range $[0,1]$, directly generated by the model as part of its response.
> > >
> > > Prior work [1] has shown that token-level probabilities, especially under chain-of-thought prompting, often fail to reflect the actual correctness of model outputs. They represent local next-token likelihoods and are not reliable indicators of global prediction confidence. In our method, by conditioning the score on the full reasoning trajectory and context, this approach helps mitigate the discrepancy between token-level likelihoods and true answer reliability.
> > >
> > > Moreover, extracting logits requires full model deployment, which is infeasible in typical scientific research settings. LGBO uses Intern-S1, a 235B parameter model, far beyond the capacity of most non-ML labs. As LGBO is designed for scientific discovery—often in physics and chemistry—we prioritize usability through lightweight, API-based confidence generation that fits real-world constraints.
> > >
> > > We hope this clarifies our design choice and are happy to provide further details if needed.
> > >
> > > [1] Geng J, Cai F, Wang Y, et al. A survey of confidence estimation and calibration in large language models[C]//Proceedings of the 2024 Conference of the North American Chapter of the Association for Computational Linguistics: Human Language Technologies (Volume 1: Long Papers). 2024: 6577-6595.

---

### Author Response · Authors · 2025-11-20
**Global Rebuttal**

We sincerely thank all reviewers for their professional and thoughtful evaluation of our work.

Unlike existing approaches, LGBO uniquely incorporates LLM-generated textual guidance into the preference-based optimization loop, enabling both interpretability and strong empirical performance. Our method produces human-interpretable textual guidance throughout the optimization process. This design makes LGBO particularly suitable for scientific discovery tasks, where transparency, controllability, and expert interpretability are critical, while consistently demonstrating strong empirical performance across our benchmark tasks.

We are grateful that several reviewers acknowledged the contribution of our work. We also deeply appreciate the constructive feedback from other reviewers regarding the rigor of our theoretical presentation and the design of experimental baselines—these suggestions have substantially improved the overall quality of our paper.

In the revised manuscript, we have addressed all reviewer comments point-by-point, with key changes highlighted in blue. To address the reviewers’ concerns, we have made the following major improvements:

**New Experiments:**

- **New Baselines Added:**

  - We introduced **colaLLM**, a LLM-driven variant of colaBO. It adopts the same LLM warm-up strategy and preference prompting template as LGBO to ensure a fair comparison (in response to reviewers W7ee and M7Dg).

  - We  included **CAKE**, A method that accelerates Bayesian optimization by designing kernel functions informed by large language models.(in response to reviewer oNUD).

  - We also included **BOPRO**, an existing Bayesian optimization approach that leverages in-context learning with LLMs and serves as a strong representative of LLM-based BO methods (in response to reviewer oNUD (in response to reviewer oNUD).



  Across all evaluated benchmarks, LGBO consistently outperforms among **colaLLM**, **CAKE** and **BOPRO**, demonstrating its robust performance and adaptability across diverse optimization scenarios.

- **New Experiments Added**:

  We introduced a challenging high-dimensional experiment on the **COF** datasets, which involves **14** optimization variables (in response to reviewer oNUD). The COF dataset focuses on xenon/krypton separation, with each material represented by a 14-dimensional feature vector encoding pore geometry, textural properties, and elemental composition.  Even in this high-dimensional setting, known to be difficult for conventional Bayesian optimization, LGBO maintains leading performance, demonstrating strong scalability.

**Writing and Theoretical Clarifications:**

- We thoroughly refined the theoretical guarantees in Section 3.4 to make the presentation self-contained and to explicitly state the assumptions underlying our analysis (in response to reviewer oNUD). We believe these revisions substantially improve the theoretical clarity and rigor of the work.
- We elaborated on the design rationale and adaptive behavior of the guidance strength parameter $\lambda$, clarifying its role in controlling the influence of LLM guidance throughout optimization (in response to reviewers oNUD and 935h).
- We corrected several minor errors and revised ambiguous phrasings for clarity (in response to reviewer W7ee).

We are confident that the improvements and clarifications provided below sufficiently address the reviewers’ concerns. Should there be any remaining questions or uncertainties, we would be pleased to provide further clarification.

---

### Author Response · Authors · 2025-12-02
**Review Summary for Area Chair**

We fully understand the additional workload resulting from the recent OpenReview issue and sincerely appreciate the Program Committee’s ongoing efforts to uphold the integrity of the ICLR review process. We are deeply grateful for the dedication that you and the reviewers have demonstrated throughout this period. Below, we provide key clarifications and improvements made in response to their suggestions.

**PAPER OVERVIEW AND CONTRIBUTION**

The paper introduces LLM-Guided Bayesian Optimization (LGBO), a novel framework that continuously integrates LLM-generated semantic preferences into the Bayesian Optimization loop through a "region-lifted preference" mechanism. The key innovation involves stably shifting the GP surrogate mean based on LLM guidance while preserving the covariance structure, enabling accelerated convergence with theoretical safety guarantees.

**REVIEWER ASSESSMENTS AND AUTHOR RESPONSES**

Several reviewers agreed on the novelty and practicality of LGBO, emphasizing its elegant integration of LLM-guided semantic preferences into Bayesian Optimization via the region-lifted preference mechanism. The method improves optimization efficiency while maintaining theoretical soundness and interpretability.

At the same time, some reviewers raised concerns regarding the clarity of the theoretical exposition and the experimental setup. In response, we revised the theoretical analysis and introduced more challenging high-dimensional benchmarks and additional baselines to strengthen the paper’s rigor and empirical support.

Ultimately, reviewers acknowledged that their key concerns were satisfactorily addressed and expressed a positive overall evaluation. Below, we summarize their main points and our corresponding responses.

***Reviewer 935h***

Initial Concerns: Over-claimed interpretation of regret bounds; missing technical details regarding guidance strength parameters and implementation

Author Response: Clarified theoretical assumptions; detailed λ calibration method (λ = c/√(aᵀΣa)); added implementation specifics

Reviewer’s comment to the rebuttal: KEEP THE SCORE 6

***Reviewer W7ee***

Initial Concerns: Insufficient baseline comparisons; unverified scalability

Author Response: Added colaLLM, CAKE, BOPRO baselines; introduced 14-dimensional COF benchmark

Reviewer’s comment to the rebuttal: Don’t have time to answer to the author’s rebuttal yet and KEEP THE SCORE 8

***Reviewer M7Dg***

Initial Concerns: Theoretical clarity issues; insufficient emphasis on novelty; baseline limitations

Author Response: Revised theory section; highlighted novelty of continuous LLM integration; expanded baseline coverage

Reviewer’s comment to the rebuttal:  “I appreciate the authors' effort to modify the paper and improve the presentation. **I would like to increase my score** because 1) My major concern about the presentation seems to be addressed 2) the authors confirmed that their idea is novel, then I acknowledge this contribution.” This reply was posted on **25 Nov 2025**. RAISE THE SCORE TO 4, with soundness, presentation, and contribution all at 3.

***Reviewer oNUD***

 Initial Concerns: Low-dimensional experiments; inadequate baselines; region format simplicity

 Author Response: Added high-dimensional experiments; included additional baselines; justified region design choices

 Reviewer’s comment to the rebuttal: “I have read the rebuttal, and the authors have satisfactorily addressed my concerns. I appreciate their efforts in clarifying parts of the paper and conducting additional experiments within the stipulated time. **I am happy to increase my score**.” This reply was posted on **21 Nov 2025**. RAISE THE SCORE TO 6

 **REVISIONS SUMMARY**

We made substantial improvements across all critical areas:

Theoretical Rigor: Complete revision of Section 3.4 with clarified assumptions and corrected notation

Experimental Validation: Added multiple new baselines and high-dimensional benchmarks

Methodological Transparency: Detailed implementation specifics for guidance strength calibration and region discretization

Writing Quality: Corrected ambiguous phrasing throughout manuscript

**REBUTTAL OUTCOME HIGHLIGHT**

Reviewers generally indicated that our rebuttal effectively addressed their core concerns. Reviewer 935h was satisfied with the improved theoretical clarity. Reviewer M7Dg appreciated the strengthened arguments and acknowledged that their main concerns had been resolved. Reviewer oNUD highlighted that the added baselines, high-dimensional experiments, and lift-strength calibration addressed their feedback and improved their evaluation.
﻿
Overall, the discussion showed a clear convergence: the theoretical framing was clarified, the empirical evaluation deemed sufficient, and the design rationale behind LGBO broadly accepted. This alignment provides important context for interpreting the current reviewer positions.

---

### Meta-Review · Area_Chair_VgkD · 2025-12-09

**Summary:**

The paper describes a new approach called LGBO for integrating LLM prior knowledge at *every step* of Bayesian optimization.  The reviewers expressed the following concerns:

1. missing baselines in the empirical evaluation
2. low dimensional experiments
3. over claim of Theorem 1
4. theoretical clarity issues

**Reviewer Concerns:**

All concerns have been addressed by the rebuttal and the revised paper.  New baselines have been added to the experiments.  A higher dimensional experiment (14 dimensions) was added to the appendix.  The paper was revised to clarify that Theorem 1 only applies to a reduced setting where the LLM prior does not change.  Finally, the theory was clarified.

**Reviewer Scores:**

I expect reviewers 935h and W7ee to keep their positiive scores since the rebuttal addressed their comments.

Reviewers oNUD and M7Dg indicated that they are satisfied by the responses and wished to increase their scores.

Overall, this paper makes a good contribution to the field by demonstrating that the continual LLM feedback in a Bayesian optimization loop can spped up AI for science.  The reviewers concerns were all addressed.  The work is novel and advances the state of the art.

---

### Decision · Program_Chairs · 2026-01-26

Accept (Poster)